# Important role of stratospheric injection height for the distribution and radiative forcing of smoke aerosol from the 2019/2020 Australian wildfires

Bernd Heinold[1], Holger Baars[1], Boris Barja[2], Matthew Christensen[3,4], Anne Kubin[1], Kevin Ohneiser[1], Kerstin Schepanski[1,5], Nick Schutgens[6], Fabian Senf[1], Roland Schrödner[1], Diego Villanueva[1,7], and Ina Tegen[1]

[1] Leibniz Institute for Tropospheric Research, Permoserstr. 15, 04318 Leipzig, Germany
[2] Department of Mathematics and Physics, University of Magallanes, Avenida Bulnes 01855 Punta Arenas, Chile
[3] Atmospheric, Oceanic and Planetary Physics, University of Oxford, Oxford OX1 3PU, United Kingdom
[4] Now at Atmospheric Sciences and Global Change Division, Pacific Northwest National Laboratory, Richland, WA, 99352, USA
[5] Now at Institute of Meteorology, Freie Universität Berlin, Carl-Heinrich-Becker-Weg 6-10, 12165 Berlin, Germany
[6] Department of Earth Science, Vrije Universiteit Amsterdam, 1081 HV Amsterdam, the Netherlands
[7] Now at Leipzig Institute for Meteorology, University of Leipzig, Germany

*Correspondence to*: Bernd Heinold (bernd.heinold@tropos.de)

**Abstract.** More than 1 Tg smoke aerosol was emitted into the atmosphere by the exceptional 2019-2020 Southeast Australian wildfires. Triggered by the extreme fire heat, several deep pyroconvective events carried the smoke directly into the stratosphere. Once there, smoke aerosol remained airborne considerably longer than in lower atmospheric layers. The thick plumes traveled eastward thereby being distributed across the high and mid-latitude Southern Hemisphere enhancing the atmospheric opacity. Due to the increased atmospheric lifetime of the smoke plume its radiative effect increased compared to smoke that remains lower altitudes. Global models describing aerosol-climate impacts lack adequate descriptions of the emission height of aerosols from intense wildfires. Here, we demonstrate by combination of aerosol-climate modeling and lidar observations the importance of the representation of those high-altitude fire smoke layers for estimating the atmospheric energy budget. Through observation-based input to the simulations, the Australian wildfire emissions by pyroconvection are explicitly prescribed to the lower stratosphere in different scenarios. Based on our simulations, the 2019-2020 Australian fires caused a significant top-of-atmosphere hemispheric instantaneous direct radiative forcing signal that reached a magnitude comparable to the radiative forcing induced by anthropogenic absorbing aerosol. Up to +0.50 W m$^{-2}$ instantaneous direct radiative forcing was modeled at top of the atmosphere, averaged for the Southern Hemisphere (+0.25 W m$^{-2}$ globally) for January to March 2020 under all-sky conditions. At the surface, on the other hand, an instantaneous solar radiative forcing of up to -0.81 W m$^{-2}$ was found for clear-sky conditions, with the respective estimates depending on the model configuration and subject to the model uncertainties in the smoke optical properties. Since extreme wildfires are expected to occur more frequently in the rapidly changing climate, our findings suggest that high-altitude wildfire plumes must be adequately considered in climate projections in order to obtain reasonable estimates of atmospheric energy budget changes.

## 1 Introduction

During the record Australian 2019-2020 wildfire season, the aerosol load increased substantially over large parts of mid and high latitudes of the Southern Hemisphere due to the massive amounts of smoke aerosol injected into the stratosphere. The austral summer of 2019-2020 is remembered as Australia's Black Summer due to the unprecedented intensity and scale of wildfires. The devastating impact on local nature and life was particularly evident in the significant destruction of habitat for hundreds of endemic species (Ward et al., 2020; Wintle et al., 2020). In addition, the interactions of the fire plume with large-scale weather (Kablick et al., 2020; Khaykin et al., 2020) make the Black Summer fires also a distinct example for studying the climate impacts of stratospheric smoke injection.

Between September 2019 and January 2020, almost twice the area burnt compared to any previous record fire in Australia, emitting unprecedented amounts of smoke aerosol (Boer et al., 2020; Morgan et al., 2020) (Fig. 1a). Peaking between 29 December 2019 and 4 January 2020, the fires caused a significant input of aerosol into the stratosphere. Several intense pyroconvective towers carried this aerosol directly up to 14–16 km height in the lower stratosphere (Kablick et al., 2020; Ohneiser et al., 2020; Boone et al., 2020). The mass of smoke emitted into the stratosphere by these fires has been estimated to range from 0.6 Tg (Khaykin et al., 2020) to 2.1 Tg (Hirsch and Koren, 2021). In a slightly later publication, Peterson et al. (2021) estimate the stratospheric injection of Australian smoke in a first phase of massive pyroconvective activity from 29 to 31 December 2019 to amount 0.2 – 0.8 Tg and 0.1–0.3 Tg in the second phase on 4 January 2020. Within days, the smoke was distributed zonally across the southern mid and high latitudes, according to satellite measurements by NASA's Moderate Resolution Imaging Spectroradiometer (MODIS) (Hirsch and Koren, 2021), Cloud-Aerosol Lidar and Infrared Pathfinder Satellite Observations (CALIPSO) data (Kablick et al., 2020) as well as observations from the Stratospheric Aerosol and Gas Experiment (SAGE) III and TROPOspheric Monitoring Instrument (TROPOMI) (Khaykin et al., 2020). Furthermore, also ground-based lidar measurements at the southern tip of South America clearly showed the elevated smoke layer (Ohneiser et al., 2020). As a result, atmospheric opacity in the southern hemisphere was considerably enhanced. Between the latitudes 20°S–60°S, the total column aerosol optical thickness (AOT) at 630 nm wavelength increased to 0.16 on average in January 2020, a 51% deviation from the long-term mean, as shown for example by the observations of the Advanced Very High Resolution Radiometer (AVHRR) satellite instrument. Figure 1b shows the hemispheric dispersal of the Australian wildfire smoke in January 2020 in terms of the AVHRR AOT anomaly.

Satellite observations and global aerosol-climate model results show that this had significant effects on the radiation budget (Khaykin et al., 2020; Hirsch and Koren, 2021; Yu et al, 2021; Sellitto et al., 2022). For the stratospheric smoke from the Australian wildfires, Khaykin et al, (2020) found a cloud-free solar radiative forcing of about -1.0 W m$^{-2}$ at the top of the atmosphere (TOA) and -3.0 W m$^{-2}$ at the surface on average for the latitudes from 25°S to 60°S in February 2020, based on radiative transfer modeling using aerosol extinction profiles from the NASA Ozone Mapping and Profiler Suite Limb Profiler (OMPS-LP). Building on the method by Khaykin et al. (2020), Sellitto et al. (2022) recently provided updated estimates and conducted sensitivity analyses to examine the effects of assumed aerosol optical properties. Their best estimate for the global-equivalent cloud-free solar radiative forcing in February 2020 is -0.35 W m$^{-2}$ at TOA and -0.94 W m$^{-2}$ at surface and reaches up to -2.0 W m$^{-2}$ and -4.5 W m$^{-2}$, respectively, for the 25°S – 60°S latitude band, using optical properties characteristic for aged biomass burning aerosol. Assuming highly reflective underlying surfaces as approximation for clouds and highly absorbing particles, in contrast, they calculated a global equivalent of TOA forcing as high as +1.0 W m$^{-2}$. Hirsch and Koren (2021) derived an enhancement of outgoing solar radiation of 1.1 W m$^{-2}$ (i.e., negative forcing) in the latitude belt between 20°S and 60°S for January to March, from NASA's Clouds and the Earth's Radiant Energy System (CERES) satellite data. From model results and considering also the fast adjustment from stratospheric warming, Yu et al. (2021) obtained an estimate for global annual average clear-sky effective radiative forcing of -0.03 W m$^{-2}$ at TOA and -0.32 W m$^{-2}$ at the surface due to the smoke event.

Australia's Black Summer is among a recent series of extreme wildfires that has renewed scientific attention particularly to
wildfires with strong fire-induced convection and self-lifting. These include recent record fires in the Western United States
and Canada (2017, 2018), Siberia (2019, 2020) and the Eastern Mediterranean (2021). Triggered by the intense fire heat, the
pyroconvection can grow to pyrocumulonimbus (pyroCb) clouds which are the primary pathway of smoke injection into the
upper troposphere and lower stratosphere (Fromm et al., 2010; Fromm et al., 2019). Radiation-induced self-lifting has the
potential to cause the smoke plumes to continue rising (Boers et al., 2010). Also due to such events, biomass burning smoke
contributes considerably to the global aerosol composition, affecting the Earth's energy balance through aerosol-radiation and
tropospheric aerosol-cloud interactions (Bowman et al., 2009; Streets et al. 2009; Boucher et al., 2013). Such extreme wildfires
and associated deep pyroconvection can have similar effects as volcanic eruptions in terms of stratospheric aerosol injection
and radiative impact (Peterson et al., 2018). An important component is black carbon aerosol, which is among the strongest
warming short-lived radiative forcing agent (Bond et al., 2013; Lund et al., 2020; Thornhill et al., 2021). In addition, less-
absorbing organic carbon, precursors for sulfate and other secondary aerosols are emitted. Depending on its optical properties
and the underlying surface reflectivity, the climate impact of biomass burning aerosol can vary regionally (Jiang et al., 2016;
Bellouin et al., 2020; Brown et al., 2021) and also strongly depends on the altitude of the aerosol layer (Ban-Weiss et al.,
2012). During strong pyroCb events, radiative effects can be enhanced due to long stratospheric lifetime of aerosol. While the
high-altitude injection of wildfire plumes is yet insufficiently represented in aerosol-climate models (Paugam et al., 2016), the
recent series of extreme wildfires and their potentially increased occurrence with climate change (Jolly et al., 2015; Abazoglou
et al., 2019; Dowdy et al., 2019; Kirchmeier-Young et al., 2019) call for greater attention in global climate modeling.

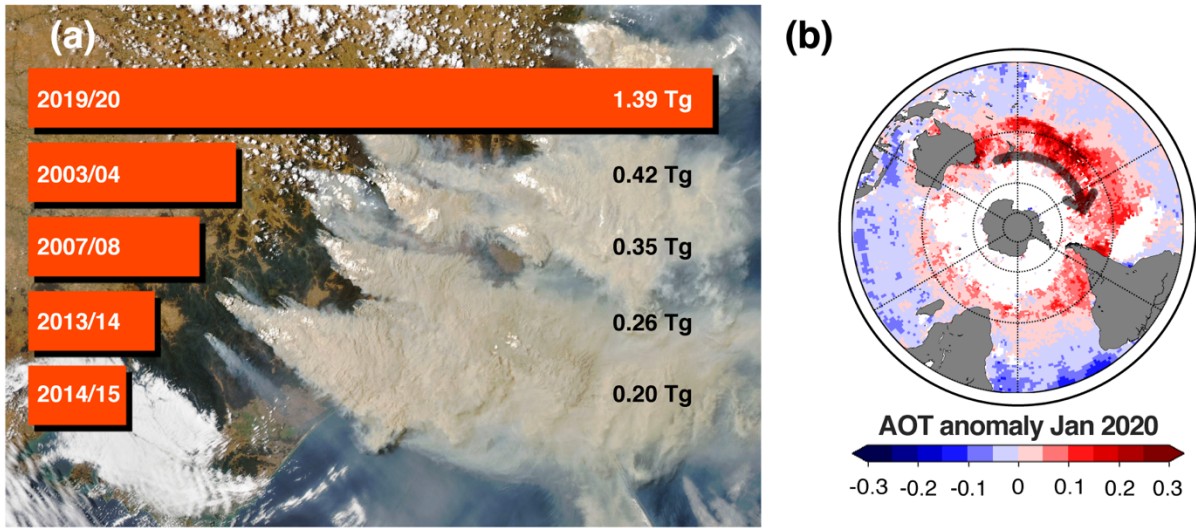

**Figure 1.** (a) Biomass burning plumes in the Canberra region in Southeast Australia as seen from NASA's Aqua satellite on
4 January 2020 (https://aqua.nasa.gov). Overlaid is a ranking of carbon aerosol emissions accumulated for the annual Southeast
Australian bushfire seasons (September to March) based on Global Fire Assimilation System (GFAS) data (Kaiser et al., 2012).
(b) Anomaly in monthly mean aerosol optical thickness (AOT) at 630 nm for January 2020 (by remapping to 1°x1°, pixels
with less than 300 valid retrievals at 0.1° original resolution are excluded, see further details in Sect. 2.3) compared to the
long-term January mean (1982 to 2019), as observed by NOAA's AVHRR instrument (Zhao et al., 2017) (missing data shown
in white, continents in gray).
In order to clarify the role of smoke injection of wildfire pyroconvection in the aerosol-climate modeling context, here we use
the aerosol-climate model ECHAM6.3-HAM2.3 (Zhang et al., 2012; Tegen et al., 2019). Specifically, we aim to show the
importance of considering these extreme fire events in determining the global energy budget, while they are not adequately
reflected in today's climate simulations. The fire emission fluxes in the model are prescribed from the Global Fire Assimilation

System (GFAS; Kaiser et al., 2012), and the injection height of Australian fire smoke is set to the tropopause level for the known pyroCb events and varied accordingly in sensitivity experiments. The modeled transport patterns are evaluated with active and passive ground-based and spaceborne remote sensing, providing the basis for analyzing the radiative impact of the carbonaceous smoke aerosol. Finally, we discuss implications and perspectives for climate models to address the extreme wildfires and their effects in a changing climate.

## 2 Observations and modeling

The analysis of the 2019-2020 Australian fire season in this study is based on global aerosol-climate simulations. An important part of the modeling is concerned with finding a configuration that best represents the pyroconvective fires. Since the typical horizontal resolution of global climate models is too coarse to explicitly resolve convection, observed pyroCb events are explicitly prescribed and the injection height of the wildfire plume is varied in terms of sensitivity experiments. Their results are compared to the original settings for biomass burning emissions as well as evaluated with ground-based and spaceborne remote sensing observations to show how realistically these can be represented if the injection heights for pyroCbs are considered accordingly. The model results are then used to investigate the impact of pyroconvective smoke injection on plume transport and radiative effects for January to March 2020.

### 2.1 AERONET sun photometer measurements

Information on column aerosol properties including aerosol optical thickness (AOT) at specific wavelengths and corresponding information on effective aerosol size are available from quality-controlled measurements by the global sunphotometer network AErosol RObotic NETwork (AERONET (Holben et al., 1998; Giles et al., 2019); http://aeronet.gsfc.nasa.gov). These data are widely used for aerosol studies including evaluation of aerosol model results. In this study we use level 1.5 or, where available, level 2 cloud-screened, 6-hour averages of AOT measurements. AERONET AOT values at 550 nm are extrapolated from the measured values at 500 nm making use of the Angstrom exponent for each observation, which in turn is computed from the ratio of observed AOT values at 500 nm and 675 nm, respectively. AOT measurements are compared to model results by linearly interpolating model values to the times and locations of the measurements of the respective AERONET stations: Punta Arenas, Chile (53.14°S, 70.89°W), Amsterdam Island (37.80°S, 77.57°E), Marambio (64.24°S, 56.63°W), Vechernaya Hill (67.66°S, 46.16°E) and South Pole (90.00°S, 70.30°E).

### 2.2 Ground-based lidar remote sensing

The lidar observations at Punta Arenas (53.14°S, 70.89°W; 9 m above sea level), Chile, were conducted in the framework of the long-term DACAPO-PESO campaign (Dynamics, Aerosol, Cloud And Precipitation Observations in the Pristine Environment of the Southern Ocean; https://dacapo.tropos.de). Main goal of DACAPO-PESO is the investigation of aerosol–cloud interaction processes in rather pristine, unpolluted marine conditions (Radenz et al., 2021). The Polly instrument (POrtabLle Lidar sYstem; Engelmann et al., 2016) was operated at the University of Magallanes (UMAG) at Punta Arenas from November 2018 until October 2021. The lidar has 13 channels and continuously measures elastic and Raman backscatter signals at the laser wavelengths of 355, 532, and 1064 nm and respective Raman backscattering wavelengths of 387 and 607 nm for nitrogen Raman scattering and 407 nm for water vapor Raman scattering (Baars et al., 2016; Baars et al., 2019). At the laser wavelengths of 355 nm and 532 nm, particle extinction coefficients, the respective extinction-to-backscatter ratio (i.e. lidar ratio), and the linear depolarization ratio are determined. Moreover, vertical profiles of the particle backscatter coefficient can be derived at these wavelengths and, additionally, at 1064 nm. The mixing ratio of water vapor to dry air is obtained from measurements in the UV. Auxiliary meteorological data, in particular temperature and pressure profiles, are required in the

lidar data analysis in order to calculate and correct for atmospheric molecular backscatter and extinction. To this end, GDAS1 (Global Data Assimilation System 1) temperature and pressure profiles with 1° horizontal resolution from the National Weather Service's National Center for Environmental Prediction (GDAS et al., 2020) were used. In addition, values for single scattering albedo at 355 nm and 532 nm were retrieved from the Polly multi-wavelength backscatter and extinction profiles from 26 January 2020 by using the Veselovskii et al. (2002) lidar data inversion (Ohneiser et al., 2022).

## 2.3 Spaceborne remote sensing

### 2.3.1 AVHRR aerosol optical thickness

Observations with the Advanced Very High Resolution Radiometer (AVHRR) onboard the National Oceanic and Atmospheric Administration (NOAA) operational satellites are available for almost four decades. For the present study, we use version 3 of the AVHRR AOT product (Zhao et al., 2017). It provides daily mean AOT at 630 nm for cloud-free pixels over none-glint water surfaces with a horizontal resolution of 0.1°. The uncertainty of a single AOT retrieval is 0.2. Because of clouds obscuring the view, there are no valid retrievals and consequently no daily averages for every pixel on every day. For the illustration of the hemispherical spread of Australian wildfire smoke in Fig. 1b, showing the observed January 2020 AOT anomaly, the original 0.1°x0.1° AVHRR data is compiled onto a grid with a spatial resolution of 1°x1° to account for sufficient samples in the temporal mean.

The long-term mean for January from 1982 to 2019 is calculated by averaging in space and time over all pixels in the 1°x1° cells. The averaging for January 2020 is performed analogously. However, due to the sparseness of the observations in a single month, 1°x1° averaging boxes with less than 300 valid retrievals in January 2020 (i.e. approx. 10% of the 100 potentially available data points per are not considered here. In particular, at high latitudes, data coverage is sparse due to the low angle of the sun and high cloud cover.

### 2.3.2 CALIOP space-based lidar observations

Lidar observations from the Cloud-Aerosol Lidar with Orthogonal Polarization (CALIOP, Winker et al., 2013) instrument are used to retrieve the extinction coefficient at 532 nm and 1064 nm. We use the level 2 version 4 aerosol profile product which is averaged over 5-km horizontal segments along the near nadir-view ground track (05kmAPro product). The cloud-aerosol discrimination (CAD) score is used to include only those columns in which at least one aerosol retrieval was successfully performed, using a threshold of < -20 CAD scores. This level of quality screening is the same as that described in Winker et al. (2013). However, despite the use of the highest quality data, CALIOP is known to frequently fail to detect thin aerosol layers in the upper troposphere and lower stratosphere (Watson-Parris et al., 2018; Liu et al., 2019). The CALIOP level 2 aerosol classification selection algorithm defines six aerosol types: clean marine, dust, polluted continental, clean continental, polluted dust, and smoke which is based on the extinction-to-backscatter ratio (i.e., lidar ratio). Comparisons of the CALIOP backscatter with airborne measurements using a High Spectral Resolution Lidar (HSRL), conducted during the ObseRvations of Aerosols above CLouds and their intEractionS (ORACLES) campaign independently demonstrated the lack of detection of these aerosol types using the CALIOP lidar, and as such, have carried out the necessary steps to account for these biases as discussed in detail in Watson-Parris et al. (2018). As a result of its low sensitivity the mean fraction of aerosol detected by CALIOP is globally up to 44% lower than the aerosol-climate model ECHAM-HAM (Watson-Parris et al., 2018). Despite this bias, the substantial increase in aerosols resulting from the Australian wildfires is evidently detected by CALIOP. While sampling and detection biases occur on individual profiles the trends in the extinction profiles offer valuable constraints for the ECHAM-HAM model.

**2.4 Aerosol-climate simulations**

**2.4.1 Model description and setup**

The simulations for this study were made with the global state-of-the-art aerosol-climate model ECHAM6.3-HAM2.3 (Tegen et al., 2019). This model uses the aerosol microphysics model M7 (Vignati et al., 2004) to predict the evolution of black carbon (BC), organic carbon (OC), sulphate, sea salt and mineral dust. The mass and number concentrations of the aerosols are influenced by emission, loss processes, particle microphysics and atmospheric chemistry. The aerosol particles can interact with radiation and clouds. Atmospheric radiative transfer in ECHAM6.3 is computed using the PSrad/RRTMG (Rapid Radiative Transfer Model for GCMs; Iacono et al., 2008; Pincus and Stevens, 2013) radiation package, which considers 16 shortwave and 14 longwave wavelength bands, respectively, as well as relevant atmospheric compounds. Aerosol optical properties are dynamically computed in the model depending on chemical composition and particle size. For each aerosol mode, the Mie-scattering parameter and the volume-averaged refractive indices are derived, assuming internal mixing, and are mapped to pre-calculated values of extinction cross section, single scattering albedo, and the asymmetry parameter from a look-up table. For analyzing the instantaneous direct aerosol radiative forcing, a double call to the radiation scheme exists to exclude atmospheric adjustments in the diagnostic.

Anthropogenic and biomass burning emissions of aerosols are prescribed. Daily data from the Global Fire Assimilation System (GFAS; Kaiser et al., 2012) based on fire radiative power observations by the MODIS instruments aboard NASA's Terra and Aqua satellites are used for the biomass burning related aerosol emissions of BC, OC, sulphate and dimethyl sulphide. In its original version, 75% of the biomass burning aerosol mass is injected in the planetary boundary layer (PBL), 17% in the first layer above the PBL and 8% in the second layer above the PBL. A more suitable representation of the smoke injection height for this extreme wildfire event is explored in a series of sensitivity experiments. In these simulations, Australian smoke aerosol is directly emitted into the tropopause region in different configurations for known days of pyroconvective activity, as described in the next Section 2.4.2.

The ECHAM6.3-HAM2.3 simulations were performed for the time period November 2019 to March 2020, using T63 horizontal resolution (approximately 1.875°x1.875°). In the vertical, the model is set up with 47 levels with increasing layer thickness from the ground to 0.01 hPa (~80 km). The vertical resolution ranges from approximately 70 m at surface to 500 m at 2.5 km and 1100 m at 15 km height and coarsens accordingly thereabove. The dynamics in all simulations was nudged towards ECMWF ERA5 reanalysis data (Hersbach et al., 2020). Sea surface temperatures and sea-ice concentrations were prescribed as lower boundary conditions using Atmospheric Model Intercomparison Project (AMIP) data (Giorgetta et al., 2012). Concentrations of long-lived greenhouse gases were specified following the Representative Concentration Pathway (RCP) 4.5 scenario. Output was written every 6 hours. The simulated aerosol properties include AOT at 550 nm and vertical profiles of aerosol extinction at 532 nm wavelength from the online lidar simulator, implemented specifically for comparisons with CALIOP and ground-based lidar measurements.

**2.4.2 Sensitivity experiments on wildfire smoke injection**

Wildfire injection heights are usually parameterized in coarser-scale models by schemes of various complexity (Paugam et al., 2016), but these do not necessarily represent the deep pyroconvection that is observed during very intense wildfires (Remy et al., 2017; Haarig et al., 2018; Ohneiser et al., 2020). In order to reconstruct the elevated smoke injection due to pyroconvection and to explore the impacts on plume transport and climate radiative effects of the 2019-2020 Australian fire plume, we adapted the high-altitude smoke injection height by pyroconvection for the days 29 – 31 December 2019 and 4 January 2020 (pyroCb days) in the model, on which strong pyroconvective activities were reported in the Southeastern Australian region affected by the fire (Kablick et al., 2020). Since no direct information was available on the actual pyroconvective injection heights, these

were varied in the model in sensitivity experiments and verified with the range of above-mentioned remote sensing observations, in particular with the lidar measurements over Punta Arenas in Chile.

The Australian fire emissions in the model, based on the GFAS inventory, are 0.6 Tg and 0.2 Tg for the two pyroconvective phases 29 – 31 December 2019 and 4 January 2020, respectively, with a black carbon to total carbon (BC/(BC+OC)) ratio in the fire emissions of about 0.06 – 0.07. These emission values agree well with the previously mentioned estimates by Peterson et al. (2021) and are kept unchanged throughout the sensitivity experiments. For the four pyroCb days (29 – 31 December 2019 and 4 January 2020), the smoke injection from Southeastern Australia was set to the model layers above and below the tropopause as in the scenarios listed in the following: '*TP+1*': 100% smoke injection into the model layer above the tropopause; '*TP*': 100% smoke injection into the model layer containing the tropopause, '*TP-1*', 100% smoke injection into the model layer below the tropopause; '*TP1_8020*': as TP+1 but only 80% of the emitted smoke injected above the tropopause and 20% distributed between tropopause level and surface; '*TP1_5050*': as TP+1, but only 50% of the emitted smoke injected above and 50% distributed below the tropopause; and '*14km*': smoke injection into 14 km height as suggested from the spaceborne CALIOP lidar measurements showing smoke plumes in the lower stratosphere up to 17 km near the Australian continent (Hirsch and Koren, 2021). It is important to note that a larger height range is directly affected because of the model layer thickness at these heights.

In addition, a reference simulation using the original model configuration with 75% wildfire emissions within the planetary boundary layer and 25% into the two model layers above was carried out, which hereafter is referred to as *BASE* case. To estimate the input of fire aerosol to the stratosphere from the pyroconvective fires, a model run was also performed in which the Southeastern Australian wildfire emissions were set to zero for the pyroCb days while they were treated as in the original setup for all other days (referred to as case *NoEmiss*). Further experiments include model runs without interactive aerosol-radiation interaction for the *BASE, TP+1*, and *TP1_8020* case scenarios in order to test the hypothesis that self-lifting due to radiative heating has significantly influenced the smoke plume evolution. The different model experiments are summarized in Table 1.

**Table 1.** Overview of scenarios simulated with ECHAM6.3-HAM2.3 using different assumptions for the emission height of the emitted biomass burning aerosol over Southeastern Australia.

| Scenario | Description |
|---|---|
| *BASE* | Standard emission height as prescribed in the ECHAM-HAM model for wildfires (75% in PBL, 17% in the first layer and 8% in the second layer above PBL) |
| *NoEmiss* | Wildfire smoke emission set to zero for the pyroCb days 29 – 31 December 2019 and 4 January 2020 in Southeastern Australia |
| *TP+1* | Wildfire smoke emission from Southeastern Australia injected into the model layer above the tropopause for the pyroCb days |
| *TP* | As *TP+1*, but smoke injection into the model layer containing the tropopause |
| *TP-1* | As *TP+1*, but smoke injection into the model layer below the tropopause |
| *TP1_8020* | As *TP+1*, but only 80% of the emitted smoke injected above the tropopause and 20% distributed between tropopause level and surface |
| *TP1_5050* | As *TP+1*, but only 50% of the emitted smoke injected above the tropopause and 50% distributed between tropopause level and surface |
| *14km* | Wildfire smoke emission from Southeastern Australia injected into the model layer around 14 km height for the pyroCb days as suggested from satellite lidar observations |

## 3 Results

### 3.1 Smoke transport simulation and model evaluation

According to satellite observations, the 2019-2020 Australian wildfire plume considerably increased the AOT of the usually pristine Southern Hemisphere (Hirsch and Koren, 2021). The average AOT at 630 nm wavelength derived from the AVHRR satellite instrument between 20°S–60°S was increased to 0.16 for January 2020, which implies a 51% offset from the long-term mean. The ground-based AERONET observations also show significantly increased AOT values after a major peak around or after mid-January. For example, at the Punta Arenas station, the January to March 2020 average 550-nm AOT was 0.10, which is more than a factor 2 increase compared to the year 2019. One year later, in January 2021, the observed 500-nm AOT over Punta Arenas was still increased with a monthly mean of 0.06 (50% increase relative to the monthly mean AOT for January 2019).

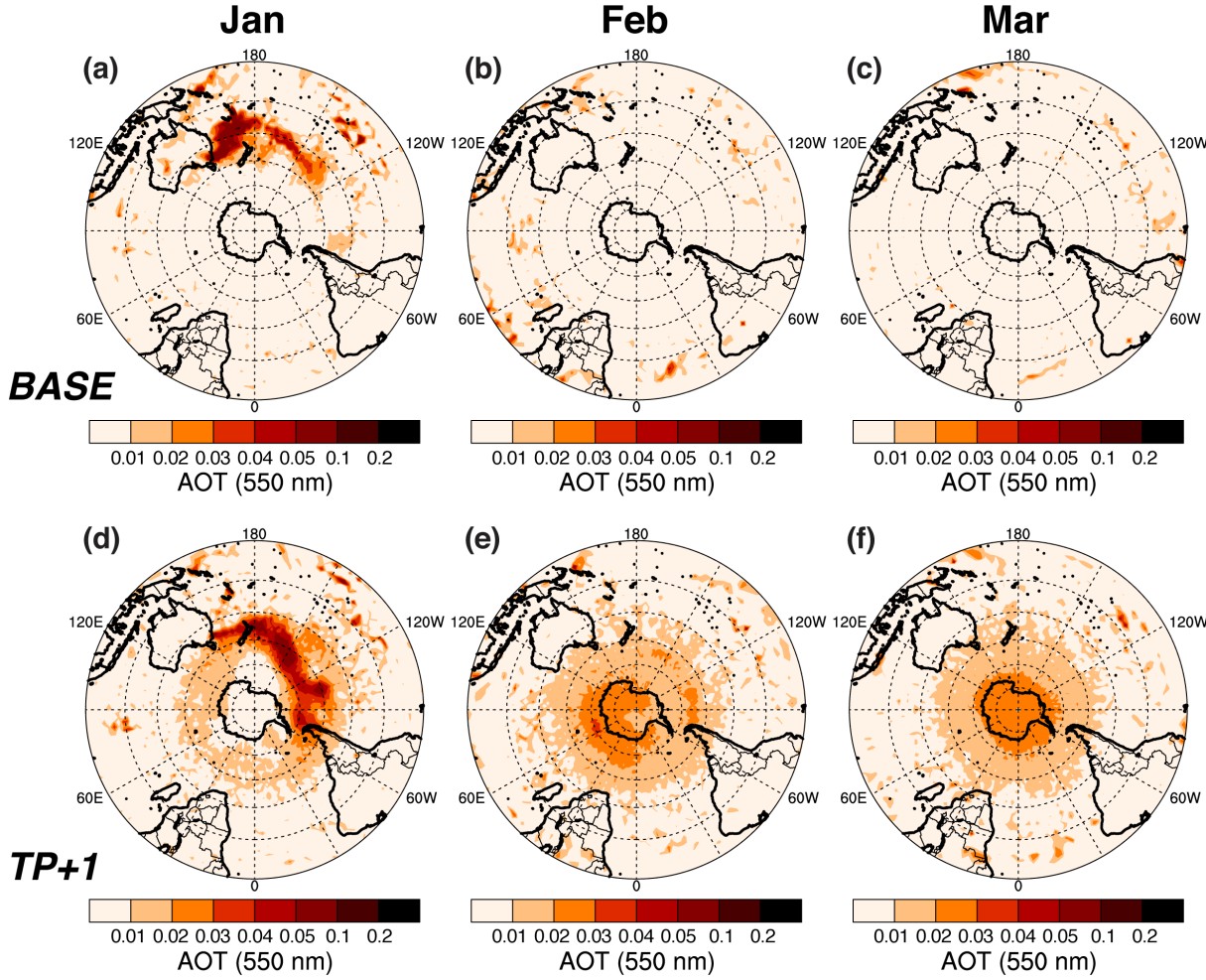

**Figure 2.** Monthly mean simulated AOT differences for January to March 2020 for the cases *BASE - NoEmiss* (top panels) showing the contribution of smoke AOT for the case when no smoke injection by pyroconvection is prescribed in the model, and *TP+1 – NoEmiss* (bottom panels), showing the effect on AOT of smoke injection into the model layer above the tropopause for the pyroCb days 29 – 31 December 2019 and 4 January, 2020 in Southeastern Australia.

The dispersal of this smoke plume is reproduced using the global aerosol-climate model ECHAM6.3-HAM2.3 with the

pyroconvective injection heights prescribed. The comparison of the modeled AOT of the *BASE* and *TP+1* experiments,

respectively, and that of the *NoEmiss* experiment (see Fig. 2) provide an insight into the AOT distribution caused by the

pyroCb events on 29 – 31 December 2019 and 4 January 2020 and illustrate the role of the smoke injection height. While the

monthly mean smoke AOT is simulated in absolute values as high as 0.26 and 0.22 for January just downwind of the fire

region in Southeast Australia for the *BASE* and *TP+1* experiments, respectively, the results of the *BASE* experiment do not

show increased smoke AOT eastward of 120°W in January and none in the later months (Fig. 2a–c). In contrast, the results of

the *TP+1* model experiment in which the smoke aerosol was injected into the model layer above the tropopause for the four

Southeastern Australian pyroCb days show persistently enhanced smoke AOT south of 30°S with AOT differences between

*TP+1* and *NoEmiss* of 0.01 to 0.03 until at least March 2020 (Fig. 2d–f). Also, a southward transport of the stratospheric smoke

during the three months leading to maximum smoke AOT anomaly above Antarctica in March is evident. Similar smoke

transport to Antarctica was reported by Jumelet et al. (2020) for the earlier major Australian fires in 2009. The effect of the

stratospheric transport of the smoke plume on simulated monthly mean AOT from the Australian wildfires is shown in Fig.

2d–f. For the simulations that consider an explicit prescription of the aerosol injection into the upper troposphere or lower

stratosphere, the model shows significantly enhanced AOT in large parts of the Southern Hemisphere. These model results

indicate, in agreement with observations (Khaykin et al., 2020; Ohneiser et al., 2020), that elevated levels of wildfire smoke

were sustained over several months and markedly impacted the radiative conditions in the Southern Hemisphere.

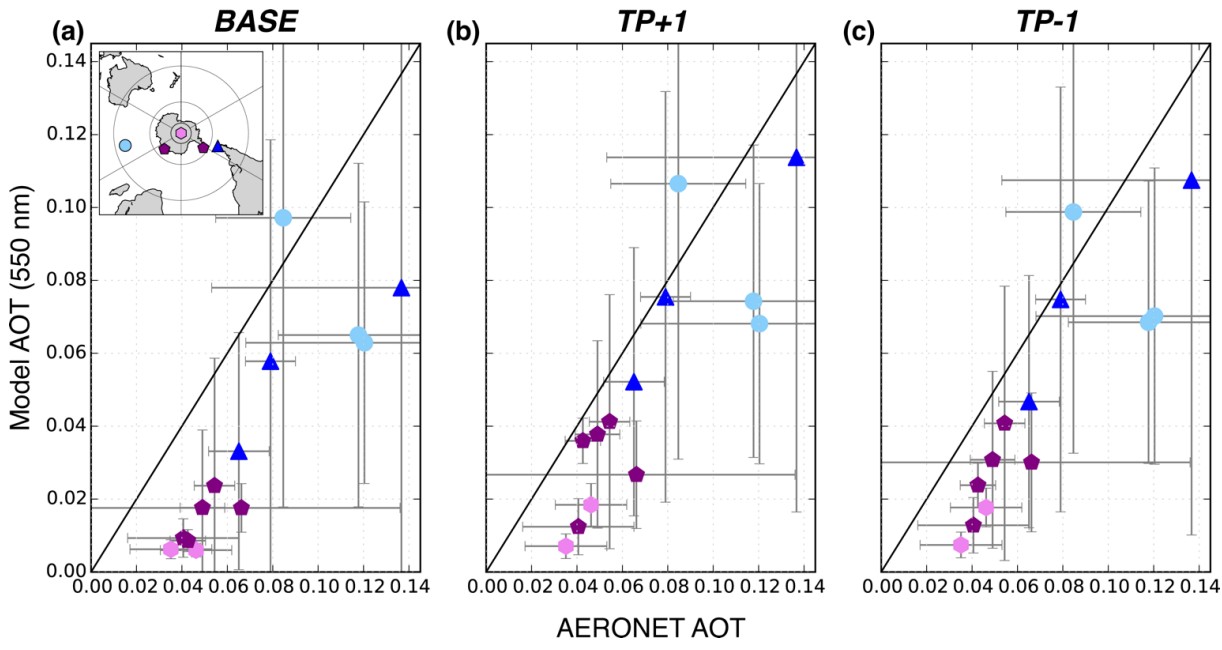

**Figure 3.** Scatter plots of observed versus simulated monthly mean 550-nm AOT at southern mid and high-latitude

AERONET stations for January to March 2020. The error bars represent the standard deviation based on daily values.

Compared are model results for the cases (a) BASE, (b) TP+1, and (c) TP-1. The stations are color-coded respectively: Punta

Arenas, Chile (53.14°S, 70.89°W), blue triangle; Amsterdam Island (37.80°S, 77.57°E), light blue circles; Antarctic Stations

Marambio (64.24°S, 56.63°W) and Vechernaya Hill (67.66°S, 46.16°E), purple pentagons; South Pole (90.00°S), light

purple hexagons.

To evaluate the representation of smoke emission height during the pyroCb days, the model results for the different sensitivity

cases representing different injection heights are compared to sunphotometer measurements of mid- and high-latitude

AERONET stations in the Southern Hemisphere (Holben et al., 1998) for the months January to March 2020 (Fig. 3), and with

ground-based lidar measurement from the PollyXT instrument at Punta Arenas for several days January 2020 (Fig. 4).

Particularly for the AERONET stations located in Antarctica the observed AOT was enhanced in early 2020 compared to previous years. The agreement of model results with AOT measured at five AERONET stations is clearly better for the cases *TP+1* and *TP-1* with prescribed fire injection heights compared to the *BASE* case (Fig. 3). Considering the overall very low levels of AOT at the Southern Hemisphere sites, the alternative injection heights lead to a substantial improvement with up to 65% higher modeled AOT values, e.g. at Punta Arenas. Still, all model results show a negative bias compared to the observations, indicating that the modeled effects of the smoke will underestimate the actual load and potentially the solar absorption of stratospheric smoke (see also Sect. 3.2). In the *BASE* case the bias is on average about 30% larger than for the other cases representing smoke injection into the upper troposphere and lower stratosphere, and the correlation is also slightly lower, at least compared to the *TP+1, TP, TP-1* and *TP1_8020* cases (Table 2). The results for *TP+1, TP, TP-1, TP1_8020* and *14km* agree similarly well with the observations, with less agreement for the *TP1_5050* case. The two cases *BASE* and *TP1_5050* therefore represent the observations worst, while no clear best fit is apparent for the other setups. The underestimation of the fire aerosol loading in all configurations can be caused by a still too low source strength in the GFAS data compared to, e.g., Peterson et al. (2021) or is partly due to missing secondary aerosol formed in the plume, which is not considered by the model.

**Table 2.** Statistical key figures for the comparison of measured and simulated AOTs for the different model cases at the AERONET sun photometer stations Punta Arenas, Chile (53.14°S, 70.89°W), Amsterdam Island (37.80°S, 77.57°E), Antarctic Stations Marambio (64.24°S, 56.63°W), Vechernaya Hill (67.66°S, 46.16°E), and South Pole (90.00°S). The numbers in bold denote the case with the best match for the respective statistical variable, the number in brackets the case with least agreement (excluding case *BASE*). From top: Normalized Root Mean Square error (NRMS; normalized by mean), bias, Pearson correlation coefficient (R), and probability of correlation (p-value).

| | *BASE* | *TP+1* | *TP* | *TP-1* | *TP1_8020* | *TP1_5050* | *14km* |
|---|---|---|---|---|---|---|---|
| NRMS | 0.61 | **0.43** | 0.44 | 0.45 | 0.47 | (0.51) | 0.47 |
| Bias | -0.035 | **-0.021** | **-0.021** | -0.024 | -0.025 | (-0.028) | -0.025 |
| Correlation R | 0.84 | 0.84 | 0.86 | **0.87** | 0.86 | 0.84 | 0.84 |
| p-value of Correlation | $3 \times 10^{-4}$ | $3 \times 10^{-4}$ | $1 \times 10^{-4}$ | $1 \times 10^{-4}$ | $2 \times 10^{-4}$ | $3 \times 10^{-4}$ | $3 \times 10^{-4}$ |

The Australian wildfire smoke was observed in early 2020 above Punta Arenas with a ground-based lidar. Pronounced smoke layers arrived first on 8 January and were clearly above the local tropopause (Fig. 5). The altitude of the observed smoke plumes steadily increased and reached top heights of 26–27 km at the end of January. For four observations in January and February 2020 (Fig. 4), the exceptionally thick smoke plume is also evident in the measured extinction coefficients. This remarkable wildfire smoke layering in terms of structure and magnitude shown in Fig. 4 can only be captured by the model with stratospheric Australian fire injection heights. Although the grid-cell to point-measurement comparison remains uncertain in detail, again a tendency for an underestimation of the stratospheric smoke is apparent in Fig. 4. This could be due to an underestimation of the fire emissions or partly due to missing secondary smoke aerosol, which is not included in the model, as already suspected from the AOT comparison.

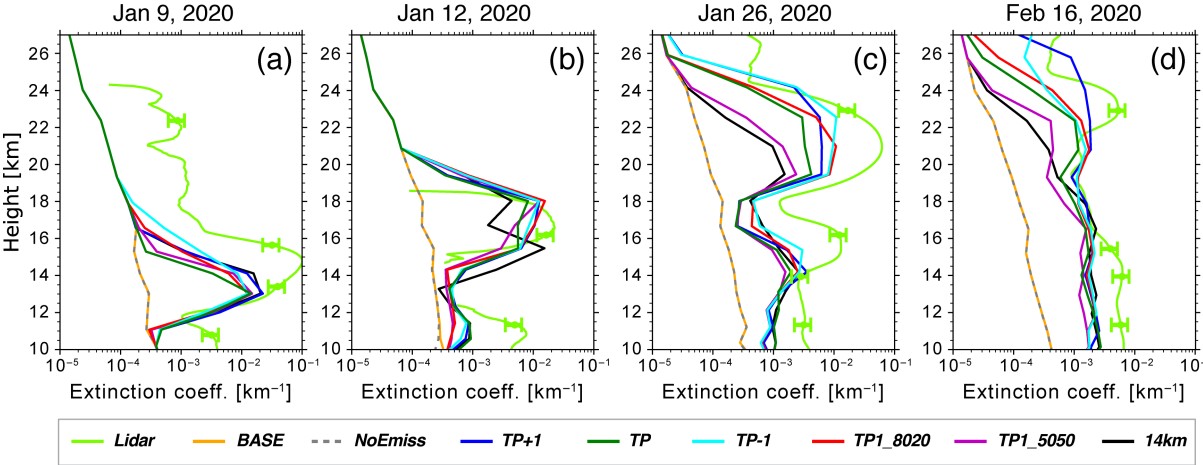

Figure 4. Comparison of modeled and observed profiles of aerosol extinction coefficients at the field site in Punta Arenas for 9, 12, and 26 January and 16 February 2020. Error bars indicate the estimated lidar measurement uncertainties of 30%, values below the lidar detection limit are omitted from the graph. See Table 1 for details of the different simulations.

When using the model with original injection height (*BASE* case), none of the structures in the stratosphere can be simulated, just like in the NoEmiss case. This gives the clear evidence that the deep pyroconvection in the wildfire hotspots in Southeast Australia did emit smoke well above the usually assumed injection heights (Remy et al., 2017; Val Martin et al., 2018). The model results also indicate the role of absorptive aerosol heating for the vertical transport of the smoke layer. In the lidar profiles, a continuous rise of the smoke layer is visible, with plume center heights increasing from 15 km to 23 km (Figs. 4, 5). The radiatively-induced self-lifting of smoke can only be reproduced if aerosol-radiation interactions are considered in the simulations that finally lead to a considerable absorptive heating and associated buoyancy production (see discussion below, including Fig. 6).

The role of the self-lifting of the smoke caused by the radiative heating of the absorbing BC aerosol in the smoke is also illustrated by the vertical distribution of modeled BC mixing ratios (shown for case *TP1_8020* in Fig. 6) averaged for January 2020 at 35°S latitude where the fires occurred. The BC mixing ratios for a model simulation where the aerosol is not interacting with radiation and thus do not heat the smoke containing atmospheric layers the smoke BC remains below 20 km height, while ascending to 24 km for radiatively interactive aerosol in the model (Fig. 6a, b). The monthly heating rate increase caused by the wildfire smoke leading to the self-lifting of the smoke plume is computed as the difference between the *TP1_8020* and the *NoEmiss* scenarios (Fig. 6 c). This heating rate reached monthly mean shortwave values up to 1.7 K day$^{-1}$ in January 2020.

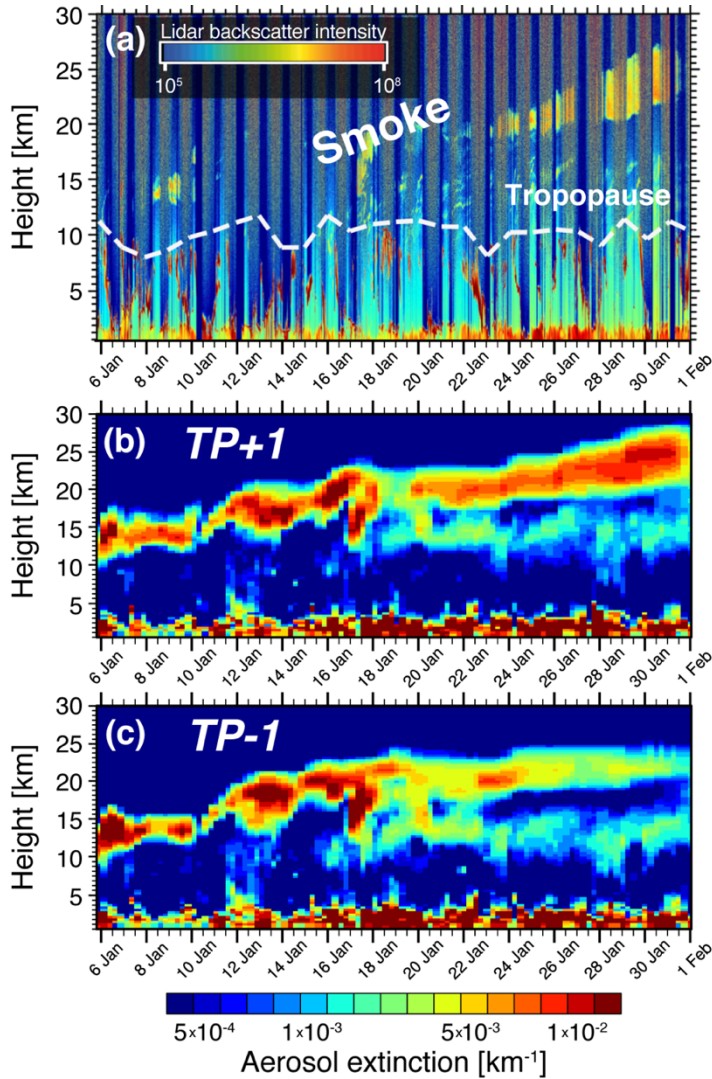

**Figure 5.** Comparison of the pattern of the temporal evolution of stratospheric smoke observed by lidar measurements and

model results at Punta Arenas, Chile for January 2020. (a) Time-height curtain plot of aerosol attenuated backscatter coefficient

from the PollyXT lidar at Punta Arenas in southern Chile (53.14°S, 70.89°W). (b, c) Simulated aerosol extinction for the model

results for the cases *TP+1* and *TP-1*, respectively.

Figure 5 qualitatively compares the development of the smoke extinction profile for the cases *TP+1* and *TP-1* with the aerosol

backscatter measurements at Punta Arenas, where the rise of the smoke plume center to 24 km by 31 January is particularly

well matched for the *TP+1* case (cf. Fig. 5a and 5b). For the other model scenarios with prescribed pyroCb smoke injection,

the plume is lifted to lower heights of 20-21 km by the end of January. But even for the case *TP-1* for which the smoke was

injected below the tropopause the smoke has lifted into the stratosphere in the model (Fig. 5c). This result underlines the

importance of a correct representation of fire injection heights, especially for intense wildfires, which is essential to realistically

assess the radiative effects of smoke plumes.

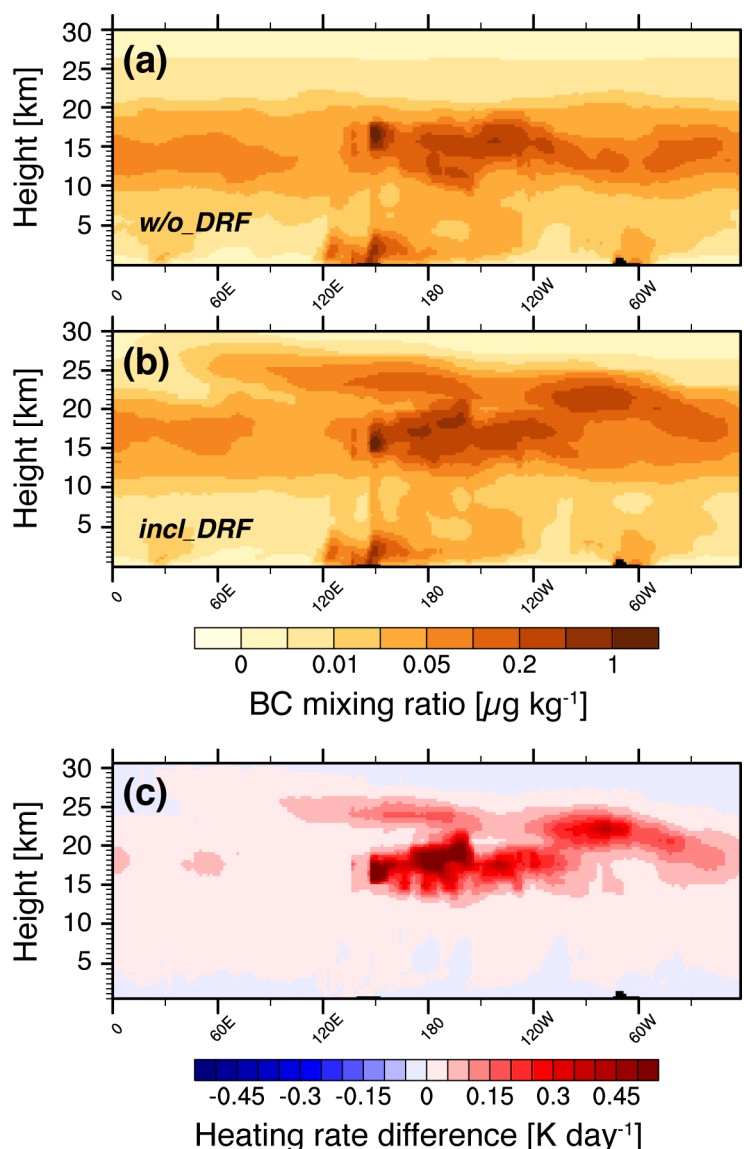

Figure 6. Longitude-height distributions of modeled black carbon mixing ratios at latitude 35 °S on average for January 2020 for the case *TP1_8020* (a) without (*w/o_DRF*) and (b) including direct aerosol radiative forcing (*incl_DRF*). (c) Change in monthly mean *shortwave* heating rate caused by the absorption of solar radiation by the Australian wildfire smoke from the pyroCb days, computed as the difference between the cases *TP1_8020* and *NoEmiss*.

Evidence that the 2019-2020 Australian wildfires caused significantly increased upper tropospheric/lower stratospheric aerosol loading across the Southern Hemisphere is also shown by the CALIOP satellite lidar observations in Figure 7. The extinction profiles averaged over the latitudes 30ºS–60ºS and the region between Australia and South America (145°E–70°W) are considerably enhanced up to 12 km altitude for November to December 2019 compared to those of the previous years 2016 to 2018. In the period from January to March 2020, the extinction is again massively increased in the altitude range from 8 km to 24 km with a peak at an altitude of 15 km. Interesting to note is also the fact that the CALIOP profile for January to March 2020 (brown line in Fig. 7) comprises about 50 times as many retrievals in the upper troposphere and lower stratosphere as the other averaging periods. This difference in sampling with far more CALIOP aerosol detections is clearly a response to the Australian wildfires. Comparing the model results with the CALIOP observations, it can be seen that our approach of prescribing pyroconvective smoke injection also reproduces well the vertical distribution of Australian wildfire aerosol across the southern mid-latitudes between Australia and South America, as shown in example of scenario *TP1_8020*. Discrepancies at altitudes above 15 km are likely related to the CALIOP sampling bias discussed in Sect. 2.3 while between 4 km and 12 km the model underestimates the fire aerosol likely because of the smoke injection mainly in the tropopause region, which is also

1   to be seen in Fig. 4. In the boundary layer, a slight overestimation occurs again in the model. Overall, these misrepresentations

2   are probably due to the comparatively simple approach to simulate the deep pyroconvective smoke injection.

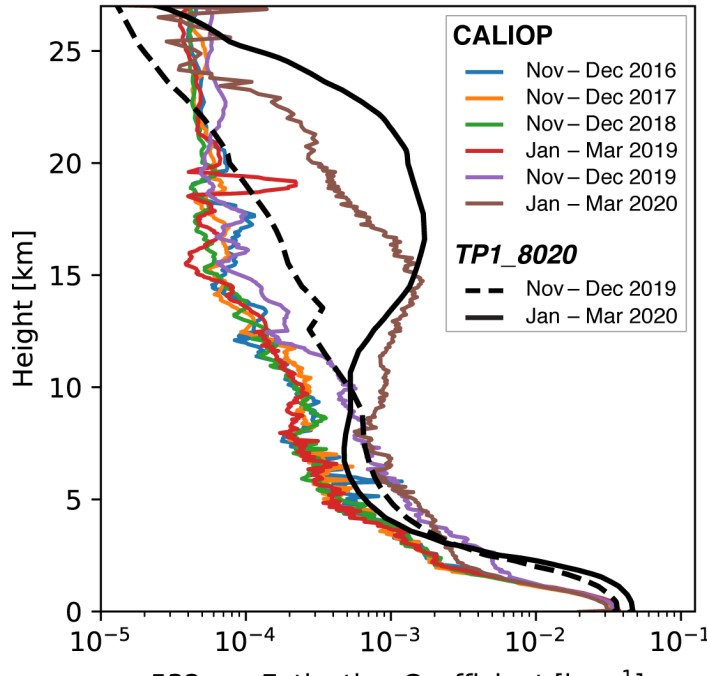

**Figure 7.** Mean vertical profile of the 532-nm retrieved extinction coefficient from CALIOP (colored lines) for several selected

periods within 2016 – 2020 and the 2019-2020 results from the ECHAM6.3-HAM2.3 model for the *TP1_8020* case, averaged

over the area covering the latitudes and longitudes between 30ºS and 60ºS and 145°E and 70°W.

**3.2 Estimates of direct radiative perturbation**

Regionally varying climate forcing agents such as aerosols substantially modulate the greenhouse forcing by anthropogenic

aerosol particles and gases. We find that the individual extreme Australian fire event caused a significant direct instantaneous

shortwave radiative forcing signal as shown in Fig. 8. The instantaneous direct aerosol radiative forcing in the model is

calculated by calling the radiation scheme twice in each simulation in order to diagnose the radiative forcing without affecting

the atmospheric conditions such as dynamics, moisture fields and clouds. The instantaneous shortwave forcing due to the

elevated smoke layers ranged up to +0.50 W m$^{-2}$ at TOA averaged for the Southern Hemisphere for January to March 2020

under all-sky conditions for the scenario *TP+1* (Table 3). This would correspond to a global-average TOA shortwave radiative

forcing of +0.25 W m$^{-2}$. In Table 3, the range of forcing estimates is given for all considered model scenarios except the clearly

unrealistic cases *BASE* and *TP1_5050*. This instantaneous forcing by the singular fire event is of similar magnitude as the

latest multi-model estimate of the global-average instantaneous forcing for all anthropogenic black carbon with +0.28 (0.13–

0.37) W m$^{-2}$ (Thornhill et al., 2021). Previous studies, in contrast, found negative TOA radiative forcing estimates

of -1 W m$^{-2}$ (Khaykin et al., 2020) for this event but only considered clear-sky situations in which the smoke aerosols appear

brighter over the dark ocean surface due to the dominant scattering effect (Bellouin et al., 2020). However, the elevated

Australian smoke layers that contain absorbing black carbon were located above clouds and to a large extent also above the

strongly reflecting snow and ice cover of the Antarctic. Over such bright surfaces, the balance between aerosol scattering and

absorption is shifted and smoke aerosol darkens the scene seen from TOA. At surface (bottom of atmosphere, BOA), the clear-

sky instantaneous solar radiative forcing was estimated to ranging from -0.68 to -0.81 W m$^{-2}$ for the different model scenarios.

This corresponds to the short-term surface dimming caused by a large volcanic eruption (Andersson et al., 2015; Schmidt et

al., 2018). On the other hand, according to the model, the smoke-containing air layer itself experienced significant absorptive heating with maximum shortwave heating rates of 1.7 K day$^{-1}$ on average in January 2020 for the TP+1 case. While the effective TOA radiative forcing is expected to be low due to stratospheric adjustment to the instantaneous forcing, these heating rate changes may have the potential to trigger responses in the atmospheric dynamics. Khaykin et al. (2020) actually showed that a self-sustained 1000-km anticyclonic vortex formed as a result, which traveled through the stratosphere for weeks, accompanied by a local ozone reduction. Such a vortex is also seen in the 50-hPa wind fields of the ECHAM6.3-HAM2.3 simulations (not shown here). The analysis of the atmospheric dynamic effects, however, is the subject of a follow-up study.

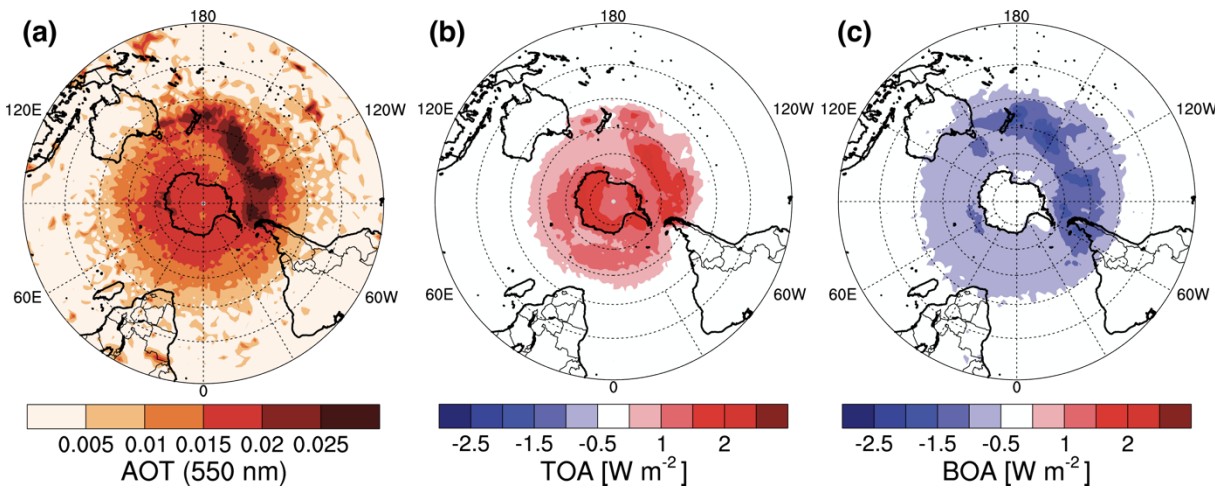

**Figure 8.** AOT and estimates of shortwave radiative forcing of the 2019-2020 Australian wildfire smoke plume in the Southern Hemisphere. Model results of (a) AOT and (b, c) instantaneous shortwave radiative forcing of the elevated smoke aerosol layer, averaged over the months January to March 2020. All values are differences between model ECHAM6.3-HAM2.3 results with Australian wildfire smoke injection for the scenario *TP+1* and *NoEmiss*. The instantaneous radiative flux differences are shown for all-sky conditions at top (TOA; b) and bottom of the atmosphere (BOA; c).

Major uncertainties in the model estimates of aerosol radiative forcing are due to uncertainties in AOT, particle angular scattering properties (asymmetry parameter) and in particular aerosol absorption that is characterized by its single scattering albedo (SSA). A recent comprehensive analysis of aircraft data indicates that model parameterizations may generally overestimate absorption by biomass burning aerosol due to an insufficient representation of the mixing state for fire aerosol (Brown et al., 2021). In our model, at the height of maximum extinction of the smoke plume, the ratio of black to total carbon (BC/(BC+OC) mixing ratio) is approximately 0.05 – 0.08, corresponding to a particle SSA between 0.82–0.85 at 550 nm. This is within the range of other aerosol models (Bellouin et al., 2020; Brown et al., 2021). Accordingly, the model may be biased toward too strong positive forcing. On the other hand, inversion results of multispectral lidar observations in the Northern Hemisphere from the strong 2017 Canadian fires yield an SSA of 0.80 for the stratospheric smoke (Haarig et al., 2018). For the 2019-2020 Australian fires, the lidar inversion method of Veselovskii et al. (2002) was applied to the Polly multiwavelength backscatter and extinction observations on 26 January 2020 to obtain values for the single scattering albedo by Ohneiser et al. (2022). They show an SSA of 0.79 with an uncertainty range of 0.05 for the smoke-filled vortex on 26 January above Punta Arenas in Chile, which is considered representative for the mid and high latitudes in the Southern Hemisphere (Ohneiser et al., 2022). A steady decline in the lidar depolarization ratio at Punta Arenas from around 20 % to 10 % by mid-February and below as 5 % after the end of February indicates an aging of the fire plume. However, high lidar ratios at 532 nm were observed with values ranging from 75 to 115 sr (mean 91 sr) well beyond that time, further indicating strongly absorbing smoke particles and low SSA values (see Fig. 8 in Ohneiser et al., 2022). The lidar simulator of the model,

in comparison, provides slightly lower lidar ratios at 532 nm between 70 and 100 sr for the stratospheric smoke layer. Regarding the asymmetry parameter, it is difficult to make an evaluation because the exact morphology of the smoke particles is not known. In the model, the asymmetry parameter for the Australian smoke is about 0.6 at 550 nm, which is a typical value for wildfire aerosol (e.g., Reid et al., 2005). In terms of single scattering albedo and lidar ratio, this would imply that the model underestimates absorption by Australian stratospheric smoke at least in January 2020, where the retrieval is available, whereas it tends to overestimate it in other, weaker vegetation fires according to Brown et al. (2021). The comparison to the ground-based remote sensing retrievals provides the best available confirmation for the optical properties of the 2019-2020 Australian fire smoke during this outstanding biomass burning event in the model. However, we cannot rule out the possibility that away from the core plume, more reflective secondary organic particles made a larger contribution to the stratospheric fire aerosol. In addition, dilution impacts on smoke aging with regard to particle size and mixing state could have influenced the smoke optical properties more strongly toward less absorption (Hodshire et al., 2021; Sellitto et al., 2022). The resulting effects of this spatial and temporal variability and evolution in the optical properties of the Australian wildfire plume would not or only insufficiently be represented by the model.

**Table 3.** Shortwave instantaneous direct radiative forcing (W m$^{-2}$) of the elevated smoke plume during the 2019-2020 Australian wildfires. The estimates are calculated differences between the instantaneous shortwave irradiances of the model results including stratospheric smoke injection and the case *NoEmiss* without smoke emission from Southeastern Australia for the pyroCb days, averaged over the Southern Hemisphere. Ranges are given for the different configurations *TP+1, TP1_8020, TP, TP-1,* and *14km* (see Table 1). Shown are the differences for all-sky and clear-sky conditions at top and bottom of atmosphere (TOA, BOA) averaged for the months January to March (JFM) 2020.

|  | TOA all sky | TOA clear sky | BOA all sky | BOA clear sky |
|---|---|---|---|---|
| **January 2020** | +0.45 – +0.56 | -0.02 – -0.05 | -0.54 – -0.61 | -0.86 – -0.97 |
| **February 2020** | +0.40 – +0.57 | +0.003 – +0.07 | -0.42 – -0.51 | -0.67 – -0.84 |
| **March 2020** | +0.25 – +0.37 | -0.01 – +0.07 | -0.28 – -0.38 | -0.46 – -0.63 |
| **JFM Average** | +0.37 – +0.50 | -0.02 – +0.02 | -0.42 – -0.50 | -0.68 – -0.81 |

**4 Implications and perspectives**

In order to determine the impact of biomass burning aerosol on the global energy budget, accurately estimating emission fluxes and their injection height in the atmosphere is essential. State-of-the-science global aerosol-atmosphere models generally consider biomass burning aerosol, but still show uncertainties in the spatio-temporal distribution. In particular, large emission events like Australia's Black Summer wildfires of 2019–2020 are underrepresented.

A key uncertainty is the vertical injection of fire smoke into the atmosphere that may ultimately cause misrepresentation of the plume evolution. The results of this study show that using fire emission data from the GFAS dataset and injecting the smoke into the tropopause region for pyroCb events gives results that are reasonable, although still somewhat underestimated in the present study.

The substantial increase in stratospheric AOT in the Southern Hemisphere, and thus the perturbation of the radiative balance, from the southeastern Australian wildfire smoke from just four days of pyroconvection events is remarkable. The local sub-

grid scale nature of fire plume rising challenges the representation in models beyond the 1-km scale, but especially in global models that do not resolve convection (Paugam et al., 2016; Veira et al., 2015). In these coarse models, the vertical distribution of fire emissions is based on climatological profiles (Val Martin et al., 2018) or prescribed by injection heights estimated from satellite retrievals of fire radiative power (Remy et al., 2017). While this is appropriate for the majority of vegetation fires, the vertical transport during deep pyroconvective events with potentially far-reaching effects is most likely underestimated due to the obstruction of satellite observations by dense pyroCb clouds (Remy et al., 2017). Adequate plume-rise parameterizations exist particularly for mesoscale chemistry-transport models, but have not found their way into climate modeling on a wider scale yet (Paugam et al., 2016; Val Martin et al., 2018; Veira et al., 2015).

Consequently, while aerosol-climate models have been shown to overestimate the radiative forcing biomass burning aerosol in general (Brown et al., 2020), they likely underestimate the wildfire aerosol impacts on the energy balance for pyro-convective fires, as the vertical location of the smoke is fundamental to its radiative impact. To solve this, adjustments are needed in the representation of biomass burning injection. By implementing a more realistic emission scenario based on aerosol-profiling observations but still using the emission fluxes from the standard GFAS database, we enhance the ability of our model to capture the extreme 2019-2020 Australian pyroCb event and can thus showcase the potential of global aerosol-climate models to realistically reproduce the spatio-temporal evolution of smoke plumes of intense wildfires. This further allows for an improved estimate on aerosol impacts on radiation and clouds. Ultimately, these improvements are essential to any estimate on the Earth's energy balance and climate state. In this respect, it is particularly important to make climate models capable of dealing with exceptional outliers of wildfires, which are anticipated to increase in frequency and severity worldwide in response to anthropogenic climate warming (Abatzoglou et al, 2019; Jolly et al., 2015; Wotton et al, 2017). The increased risk of serious wildfires is related to extreme heat and drier conditions, as well as record-low snow cover in boreal regions (Box et al., 2019; Dowdy et al., 2019). More frequent and intense fire weather extremes will also increase the likelihood of deep pyroconvection (Dowdy et al., 2019). Along with this, of course, there continues to be the challenge and need to develop better model representations of smoke aging and secondary aerosol formation, and the associated evolution of particle optical properties in the lifecycle of wildfire plumes. In essence, biomass burning emissions are an important source of aerosol particles, and individual wildfires are shown to have more widespread effects than previously assumed. An as-accurate-as-possible description, therefore, is key to successfully estimate aerosol climate effects, and future climate projections will clearly benefit from an improved aerosol representation in Earth system models.

*Code availability.* The ECHAM-HAMMOZ code is maintained and made available to the scientific community under https://redmine.hammoz.ethz.ch. The availability is regulated under the HAMMOZ Software Licence Agreement that can be downloaded from https://redmine.hammoz.ethz.ch/attachments/download/291/License_ECHAM-HAMMOZ_June2012.pdf.

*Data availability.* The ECHAM6.3-HAM2.3 model output, on which the figures are based, and the analyzed lidar profiles are accessible at Zenodo from https://zenodo.org/deposit/5571545 (Heinold et al., 2021) The ground-based Polly lidar data time series is visualized at polly.tropos.de and will become publicly available via the data portal of the European Aerosols, Clouds and Trace gases Research Infrastructure (ACTRIS) when its implementation phase is completed. Until then, raw data is available on request via polly@tropos.de. AERONET data can be obtained with the Aerosol Robotic Network download tool https://aeronet.gsfc.nasa.gov/cgibin/webtool_opera_v2_new. The Aerosol Optical Thickness CDR used in this study was acquired from NOAA's National Climatic Data Center (http://www.ncdc.noaa.gov).

*Author contribution.* BH and IT conceived the idea and led the study. BH and AK performed the model development and ran the simulations. NS provided the LIDAR simulator for ECHAM6.3-HAM2.3. IT, FS, KS, AK and BH focused on the analysis and interpretation of model results. RS analyzed NASA's AVHRR AOT climatology data, and DV researched the Earth history context. KO, HB and BB performed the lidar analysis and interpretation. BH, IT, KS, FS and RS wrote the paper with contributions from all co-authors. All authors participated in the revision and editing of the paper.

*Competing interests.* The authors declare no competing interests.

*Acknowledgements.* The authors thank Johannes Quaas and Albert Ansmann for sharing their comments on an earlier version of the manuscript. We also would like to thank the whole DACAPO-PESO team for ensuring the continuous operation of all the different instruments at Punta Arenas. Special thanks to Patric Seifert, Martin Radenz, Cristofer Jimenez and Ronny Engelmann for taking care of the lidar, measurements and ensuring high-quality data operation at the cost of spending many lifetime hours in an airplane. Igor Veselovskii is acknowledged for computing lidar inversion products. The AVHRR Aerosol Optical Thickness Climate Data Record (CDR) was originally developed by Xuepeng Zhao and colleagues for NOAA's CDR Program. We thank P. Goloub, V. E. Cachorro Revilla, P. Seifert, J. Butler, B. Holben, and A. Chaikovsky for their efforts in establishing and maintaining the AERONET sites Amsterdam Island, Marambio, Punta Arenas, South Pole, and Vechernaya Hill. The ECHAM-HAMMOZ model is developed by a consortium composed of ETH Zürich, Max Planck Institute for Meteorology, Forschungszentrum Jülich, the University of Oxford, the Finnish Meteorological Institute and the Leibniz Institute for Tropospheric Research (TROPOS) and managed by the Center for Climate Systems Modelling (C2SM) at ETH Zürich. We are grateful for computing time from the Deutsches Klimarechenzentrum (DKRZ). Computing resources at DKRZ were granted under project number bb1004. We thank Pasquale Sellitto and an anonymous reviewer for their helpful feedback.

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
