# Peer review of "Important role of stratospheric injection height for the distribution and radiative forcing of smoke aerosol from the 2019/2020 Australian wildfires"

_Atmospheric Chemistry and Physics, 2021_

## Author Comment (AC1)

**Authors' Response to Reviewers' Comments**

Manuscript No.: acp-2021-862, submitted to GMD
Title: Important role of stratospheric injection height for the distribution and radiative forcing of smoke aerosol from the 2019/2020 Australian wildfires

Authors: Bernd Heinold, Holger Baars, Boris Barja, Matthew Christensen, Anne Kubin, Kevin Ohneiser, Kerstin Schepanski, Nick Schutgens, Fabian Senf, Roland Schrödner, Diego Villanueva, and Ina Tegen

*We would like to thank the anonymous reviewer for his/her time and constructive comments, and hope that we have responded satisfactorily to all the points raised.*

**Anonymous Referee #1 (RC1)**

**General comment**

The paper presents simulations with an aerosol climate model on forest fire smoke in the stratosphere using lidar observations and the satellite based GFAS inventory. It includes several sensitivity studies on injection height since a pyro-Cb module is still not available in the global model community. The paper demonstrates the importance of wild fires for stratospheric radiative properties and is suitable for ACP after revision since it might be a valuable contribution to a hot topic in atmospheric research.

**Major comments (#MC)**

At several places clarifications are needed, including missing definitions of acronyms.
*Further explanations were added throughout the revised manuscript (see also the responses to the specific comments), and the acronyms and abbreviations are now all explained.*

Figure 7 shows large discrepancies between model results and CALIOP satellite observations concerning the vertical distribution of aerosol extinction. The figure poorly displays the lower stratosphere as the region of interest because of selection of an unphysical linear pressure coordinate. Here log(p) or altitude should be used as in the other figures. As it is, the figure gives the impression that aerosol extinction calculated by the model is severely overestimated almost everywhere, despite averaging, in contrast to the text. The difference is larger than the value mentioned in section 2.3. There appears to be something inconsistent to the results presented in the other parts of section 3.1. It might be worth, to exclude the tropics here and/or look also for other satellite data. For example, OSIRIS sees extinction peaks at 12 and 18km for January to March 2020 averaged over the southern hemisphere.
*A height axis is indeed better suited to show the relevant atmospheric altitude range. It was also found that the previous figure was incorrect due to an averaging error of the CALIOP data. This explains the impression that there was inconsistency with the other parts in Section 2.3. As suggested, we also excluded the tropics/subtropics (30°S – 60°S) and limited the longitude range to the area between Australia and Argentina (145°E – 70°W), which further improved the comparison. We have revised the figure and also added additional information to the text. Especially worth mentioning is the interesting fact that the January-*

*March 2020 period (brown line in Fig. 7) has almost an order of magnitude more aerosol retrievals in the UTLS than the averaging periods before. This is clearly a response to the Australian wildfires.*

Figure 8 would be better with satellite observations for AOT included. There are datasets of several instruments available. At least refer to Fig.1 here and use similar color bars with the same units. It looks like that the model overestimates the perturbation at Antarctica. Concerning uncertainties, it should be also taken into account that in 2019 the stratosphere was perturbed by volcanic eruptions.

*Figure 1b is shown here mainly to qualitatively illustrate the hemisphere-wide spread of the Australian smoke plume. As we also explain later in response to a specific comment, Fig. 1b and Fig. 8a are not directly comparable. Fig. 1b shows the anomaly from total long-term mean AOT. In the other hand, Fig. 8 is intended to show only the smoke AOT from the four pyroCb days.*

*The AVHRR instrument can only detect AOT over cloud-free, non-glint water surfaces. Therefore, there are no valid retrievals and subsequently daily means for each original 0.1°x0.1° pixel and day. For Fig. 1b, the original 0.1°x0.1° AVHRR data is compiled onto a grid with a spatial resolution of 1°x1° to account for sufficient samples in the temporal mean. 1°x1° pixels with less than 300 valid retrievals in January 2020 (i.e. approx. 10% of the potentially available 100 0.1°-pixels on each of the 31 days) are not considered in the spatial and temporal averaging for January 2020. This corresponds to a data coverage of approximately 10% of the 3100 potentially available retrievals for each 1°x1° grid cell (100 0.1°-pixels times 31 days).*

*In addition, we agree with the reviewer that other influencing factors, such as a stratospheric volcanic eruption, might have contributed to the AOT anomaly in Fig. 1b. This is another reason why a comparison with the model results in Fig. 8a, which only shows the AOT of the Australian smoke, is difficult. Note that no AVHRR observations are available over Antarctica, and even in January the instrument is rarely able to measure the AOT near 60°S. This information will be added in a brief manner in the figure caption and a more detailed explanation will be given in Section 2.3. The absence of robust mean observations is depicted as a white area in Fig. 1b.*

*Since the AVHRR instrument can only observe AOT over non-glint water surfaces, missing values over Antarctica, however, do not in any way imply a model overestimation, as shown by the comparison with AERONET AOTs in Fig. 3, which rather indicate an underestimation.*

**Specific comments (#SC)**

Page 1, line 19: Is the amount of injected smoke from this study or from the literature?

*This is the amount of stratospheric fire aerosol calculated by the model in this study. However, the value is also consistent with the estimates in the literature (Khaykin et al., 2020; Hirsch and Koren, 2021; Peterson et al., 2021), as we explain in detail in the Introduction (references are not allowed in the abstract).*

Page 1, line 31: Also the global value should be provided in parentheses.

*Done.*

Page 1, line 32: Provide also the value for all-sky.

*The value of +0.50 W m$^{-2}$ in the line before is actually that for all-sky conditions.*

Page 2, Figure 1 and line 10ff: Here only Fig 1a is mentioned, part 1b is mentioned first on page 6. More text is needed or the figure should be split. A definition for AOT is missing, including spellout and altitude range (Page 6 is too late). Why does AOT anomaly in Figure 1 differ from the one in Figure 8 by about a factor of 10?

*Figure 1b and the significant increase in atmospheric opacity in the southern hemisphere shown by the AOT anomaly in the AVHRR imagery is now already mentioned in the Introduction. The abbreviation AOT is now explained at this point in the text and in the caption of Figure 1. Throughout the manuscript we refer to the total column AOT.*

*As already explained in response to the main comments, the discrepancies between Fig. 1b and Fig. 8a are due to the different parameters shown. Fig. 1b shows the anomaly from the long-term average of the total AOT of the AVHRR satellite instrument while Fig. 8a shows only the AOT due to the four pyroCb days calculated from two different model scenarios. Furthermore, discrepancies result from the sampling bias of the satellite, as described above, so that local values can be significantly larger than a total monthly mean if the number of sampled days is below the total number of days in the month. In addition, other influencing factors, such as a stratospheric volcanic eruption, might have contributed to the AOT anomaly in Fig. 1b. We point this out more clearly in the text.*

Page 4, line 5: Why interpolation? The model output should be available at the time and the location of the measurements. Don't rely here on averages, especially not if the meteorology of the model is nudged to observations.

*Model results are generally available in a discrete temporal and spatial distribution. In this study, for the global aerosol-climate model ECHAM-HAM, it is a 6-hourly model output on an approx. 1.875°x1.875° (180x180 km) latitude-longitude grid. Interpolation to the much higher temporally resolved but often irregular single-point observations from AERONET is mandatory. This is common practice.*

*Nudging ensures that the simulated weather patterns are close to reality. Only then are the model results directly comparable with the measurements.*

Page 4, line 31: Vertical or horizontal resolution?

*The text says averaging along the ground track of CALIPSO, which implies a horizontal resolution. The word 'horizontal' has been added to be clearer.*

Page 4, line 36: Here something is missing. Which aerosol type? Which time? Model results need boundary conditions and cannot be a reference for observations.

*Thank you for the questions. We have added the following to the manuscript:*
*'The CALIOP level 2 aerosol classification selection algorithm defines six aerosol types: clean marine, dust, polluted continental, clean continental, polluted dust, and smoke which is based on the extinction-to-backscatter ratio (i.e., lidar ratio). Comparison of the CALIOP backscatter with airborne measurements using a High Spectral Resolution Lidar (HSRL), conducted during the ORACLES campaign independently demonstrated the lack of detection of these aerosol types using the CALIOP lidar, and as such, have carried out the necessary steps to account for these biases as discussed in detail in Watson-Parris et al. 2018.'*
*As pointed out by the authors of Watson-Parris et al. (2018), the global model ECHAM-HAM was used as a basis of reference to which they compared the CALIOP products. Of course, like any model, this one has its own uncertainties and discrepancies, but this has no impact on the conclusions in general regarding the biases of the CALIOP retrievals in the middle and upper troposphere.*

Page 5, line 12: Mention tropopause region and reason (Pyro-Cb) already here. Also it should be mentioned how many teragrams of smoke (carbon) were injected in each of the 4 events to enable comparisons with the values of other papers mentioned in the introduction (or the abstract?). More details on the relations between GFAS and estimated injected OC (organic carbon) and BC (black carbon) should be included (here or in an Appendix).

*Thanks for the suggestion. The approach of the sensitivity experiments to inject the smoke in the tropopause region for pyroCb days is now briefly mentioned in Sect. 2.4.1. We also added the amount of smoke emitted in the model during these days.*

Page 5, line 14: The 47 level version does not have a QBO and has problems with the "H2O tape recorder", i.e. the vertical transport. It is better to use L90. This should be mentioned as a possible reason for discrepancies. Is nudging applied everywhere (may cause numerical problems) or only in the troposphere and lowermost stratosphere?

*The model runs were done in nudged mode to reproduce the meteorology as closely as possible. The logarithmic ground pressure is nudged, as well as the divergence and vorticity in all model layers, with relaxation times of 24, 48 and 6 hours respectively. We are aware that this leads to bias at different levels, but only nudging allows comparability with observational data.*
*Possibly also because of the nudging we could identify the wind patterns typical for the QBO in the long model run.*
*Thank you for this advice. We will gladly take up the suggestion of a simulation with L90 for future model runs. Here, however, a rerun of the experiments is out of the scope.*
*However, it is important to emphasize that the uncertainties in the study of the Australian forest fire aerosol are mainly in the representation of the stratospheric smoke injection as we show in this paper.*

Page 6, line 2 or 4: Does this refer to the 8% mentioned on the previous page or in BASE?
*The experiments with and without interactive aerosol-radiation interaction were performed for the BASE, TP+1, and TP1_8020 case scenarios. The text was updated accordingly.*

Page 6, line 9ff: This paragraph might be better moved to the introduction. Fig. 1b is inconsistent to Fig. 2, please explain why. Or is this just a problem with the range of the colors in the figures?
*As replied to an earlier comment, Figure 1d is now already described in the Introduction.*

*The figures Fig. 1b and Figs. 2a,d are not directly comparable. Figure 1b presents AOT observations by AVHRR for January 2020. January 2020 was the month, in which the wildfire plumes where thick close to source in Southeastern Australia. In the weeks after the event, the smoke aerosol spread over a larger area and also the amount of smoke aerosol decreased. Figure 1b was therefore meant to show the apparent short-term direct effect of the wildfires and was shown only for January on purpose. In contrast, Figs. 2a,d show the differences in mean AOT for January – March 2020 between the model scenarios BASE (2a) and TP+1 (2d) against NoEmiss, respectively. Therefore, they exclusively show the contribution of the smoke-only AOT for the case where no smoke injection by pyroconvection is prescribed in the model (Fig. 2a) or for smoke injection into the model layer above the tropopause on the AOT for pyroCb days 29-31 December 2019 and 4 January 2020 in southeastern Australia (Fig. 2d). The*

*satellite-based AOT anomaly in Fig- 1b results from the difference against a long-term mean. Hence, many different aspects can contribute to the observed anomaly, e.g. wildfires in the region also beyond the 4 pyroCb days, general offset in AOT as mentioned by the reviewer in a previous general comment. Furthermore, due to the scarce observational data, the January 2020 mean of observed AOT is not directly comparable to the modeled mean AOT. For example, the AVHRR instrument can only observe AOT over cloud-free non-glint water surfaces. Therefore, there aren't valid retrievals and subsequently provided daily means for each pixel on each day. For the map in Fig. 1b, we only consider pixels with at least 5 days of valid retrievals in January 2020 to calculate the mean. This information will be added in a brief manner in the figure caption and a more detailed explanation will be given in section 2.3. The uncertainty of a single AOT retrieval is 0.2, which is in particular critical for low AOT values. The uncertainty of the January 2020 mean for a single pixel is therefore difficult to quantify. Therefore, a quantitative comparison between model and AVHRR observations was on purpose not included in the paper.*

*However, by averaging over larger areas or longer time periods this uncertainty should average out to some extent. The figure below shows the average AOT in January 2020 between 20-60°S in 15° zonal sectors from Australia westwards for both the ECHAM TP+1 simulation and the AVHRR observation. Note that here still areas with fewer valid satellite observations (more frequent in the south of the 40x15° boxes) are underrepresented in the areal mean. Further, the AOT in the figure does not correct for the different wavelengths (550nm in ECHAM, and 630 nm by AVHRR). The figure indicates the general underestimation of AOT by the model simulation that was also seen in the comparison to the AERONET observations (Fig. 3).*

Page 7, Figure 2 and line 11: Are these values out of the range of the color bar? Please adjust the color bar to accommodate this.
*Done.*

Page 8, line 17: This scenario should be also in section 2.4.1, maybe in parentheses. Or refer at least to Table 1.
*Here, the half-sentence explaining the scenario again was misleading. It is in fact the BASE case. This half-sentence was removed.*

Page 9, line 1ff: Caption too short, spell out RMS, normalized against what average(s)?
*The missing information was added to the figure caption accordingly.*

Figures 4 and 5: Standard units for extinction are "km-1", please convert axes, also to be consistent with Figure 7. I suppose the authors mean 1e-06 with Mm.
*The figures were revised accordingly.*

Figure 5: Please adjust the heights of the panels. It would be also nice to have additional panels with consistent palettes where the lidar ratio is applied for conversion.
*Figure 5 is primarily meant to give a qualitative impression of the layering of the Australian smoke plume. The quantitative comparison is already shown in Fig. 4. The height axes of the panels were unified and the color tables adjusted. However, the high temporal resolution of this lidar imagery does not allow a conversion to extinction, since a clean cloud-aerosol discrimination is not possible and thus no lidar ratio can be assumed.*

Page 11, line 8: Add "(with interaction between radiation and dynamics)"

*There seems to be a misunderstanding about the calculation of the radiative forcing. See also the reply to the comment on page 12, line 7.*
*The instantaneous forcing was actually calculated by a double call of the radiation scheme in the model, so that no dynamic influences are included in the aerosol radiative forcing and heating rates. These are in fact instantaneous estimates. Since we obviously missed to mention this, the following sentence was added to the model description: "The instantaneous direct radiative forcing from the modeled wildfire aerosol is calculated by double calling the radiation routine in ECHAM6.3-HAM2.3 in order to diagnose the radiative forcing avoiding any impact on the atmospheric conditions such as dynamics, moisture fields and clouds, but including the net solar and thermal fluxes at the bottom and top of the atmosphere."*

Page 11, line 11ff and Figure 7: As shown, the agreement is poor (not "well"). The figure has to be improved as mentioned above and more explanation is needed. The disagreement cannot be explained by sampling issues alone (section 2.3). These results are in a strong contrast to Figure 4 where the model at least follows the observed vertical patterns.
*See the reply to the corresponding main comment. Figure 7 was completely revised and additional information was added to the text.*

Page 12, line 7 and later: Taking the difference of radiative fluxes from 2 simulations is not exactly "instantaneous radiative forcing" since convection or other non-radiative processes might be different. It is, however, an estimate.
*We believe there is a misunderstanding regarding the calculation of the radiative forcing. Please note that here actually the instantaneous forcing was calculated by a double call of the radiation scheme. This means that there is no response from clouds, humidity, etc. Nevertheless, there may be negligible model deviations over the Australian area due to non-linear effects in the aerosol microphysics of the total particle population. However, we already explicitly refer to 'estimates' in the text, e.g. in the heading of Section 3.2 and therein, as well as in the Abstract and in the caption of Table 3. In addition, "estimates of solar radiative forcing" was added to caption of Fig. 8.*

Page 12, line 22: Is this number local or some kind of average?
*As mentioned earlier in the text, this is a monthly average for January 2020. The sentence was revised to be clear: "On the other hand, according to the model, the smoke-containing air layer itself experienced significant absorptive heating with maximum heating rates of 1.7 K day$^{-1}$ on average in January 2020 for the TP+1 case."*

Page 13, line 9: The particle SSA depends strongly on the partitioning between BC and OC. More information on this would be useful here, at least some typical number of the ratio with a range.
*Fair point. Together with further edits on the uncertainties of the forcing estimates in response also to Reviewer #2 we have added: "A recent comprehensive analysis of aircraft data indicates that model parameterizations may overestimate absorption by biomass burning aerosol (Brown et al., 2021). In our model, at the height of maximum extinction of the smoke plume, the ratio of black to total carbon (BC/(BC+OC) mixing ratio) is approximately 0.05 – 0.08, corresponding to a particle SSA between 0.82–0.85 at 550 nm."*

Page 13, line 14: This is in contradiction to the large high bias in Figure 7.

*Please see the reply to previous comments on Figure 7.*

**Technical corrections**

Page 16, line 19: Check abbreviation for journal.
*Corrected.*

Page 18, line 14: Please separate the 2 references.
*Done.*

Page 19, line 5: Check abbreviation for journal.
*Checked, correct as is.*

---

## Author Comment (AC2)

**Authors' Response to Reviewers' Comments**

Manuscript No.:  acp-2021-862, submitted to GMD
Title:  Important role of stratospheric injection height for the distribution and radiative forcing of smoke aerosol from the 2019/2020 Australian wildfires

Authors:  Bernd Heinold, Holger Baars, Boris Barja, Matthew Christensen, Anne Kubin, Kevin Ohneiser, Kerstin Schepanski, Nick Schutgens, Fabian Senf, Roland Schrödner, Diego Villanueva, and Ina Tegen

*We would like to thank Pasquale Sellitto for his time and constructive comments that helped to improve the submission. We hope that we have responded satisfactorily to all his points.*

**Anonymous Referee #2 (RC2)**

**General comment**

The manuscript "Important role of stratospheric injection height for the distribution and radiative forcing of smoke aerosol from the 2019/2020 Australian wildfires", Heinold et al., presents and discusses the aerosol spatiotemporal distribution and radiative forcing (RF) of ECHAM6.3-HAM2.3 model simulations of the pyro-convective paroxysmal phase of the record-breaking Australian bushfires in the fire season 2019/2020. The importance of representing pyro-convection and UTLS injection of smoke into models is discussed based on sensitivity analyses. The manuscript provides a new estimation of the RF and radiative heating of this Australian fire event, which is quite a hot topic. It certainly falls within the scopes of ACP and is very interesting for the aerosol/climate community, and is potentially suitable for publication. By the way, I have found that there are several aspects – some of them very important – that must be clarified before I can recommend publication of the manuscript. Thus, I kindly ask the Authors to reply to the following major and specific comments and to provide an amended version of the manuscript, that I would be glad to re-read when ready.

**Major comments (#MC)**

MC1) The temporal trend of the AOT obtained with ECHAM are contradictory with previously published observations. The trend shown in Fig. 2 (maximum AOT in January and then decreasing in February and March) is not consistent with, e.g., the stratospheric AOT from SAGE-III in Khaykin et al., 2020 (e.g. its Fig. 3) – paper which is cited in the present manuscript. In particular, Khaykin et al. show an increasing AOT from January to February, with a maximum in February. The Authors should discuss this marked difference. Is an increase in stratospheric-only AOT in February also present in ECHAM modelling (in Fig. 2 there's a total column AOT, I guess)? Is the possible coating of black carbon (BC) particles and evolving aerosol mixing state not well represented in ECHAM – which is quite a common feature in aerosol/climate models, as extensively discussed by Brown et al. (https://www.nature.com/articles/s41467-020-20482-9)?
*This is a central point of criticism throughout the entire review, but we have to strongly disagree with it. The argument that the temporal evolution of the smoke aerosol and*

*therefore the instantaneous radiative forcing estimates in the ECHAM-HAM simulations are supposedly misrepresented is incorrect.*

*In their figure caption of Figure 4, Khaykin et al. clearly state about the time lag between gases and aerosol particles that "The lagging increase of the aerosol mass is due to the fact that the OMPS-LP extinction retrieval saturates at extinction values above 0.01 km$^{-1}$. Profiles are, therefore, truncated below any altitude exceeding this value, which can lead to an underestimation of the early aerosol plume when it is at its thickest. This artifact, which explains the slower increase of aerosol mass than gases, persists until mid-February when the plume is sufficiently dispersed so that OMPS-LP extinction measurements no longer saturate." This clearly shows that extinction was too high to be not be measured in January and cannot be used for model comparison.*

*Peterson et al, 2021, which we now also include as a reference, provide emission strengths for the stratospheric smoke injection of 0.2-0.8 Tg for the pyroCb events on 29-31 December 2019 and 0.1-0.3 Tg for 4 January 2020. These values agree well with our simulation results. Secondary aerosol formation appears unlikely to be the explanation considering the required amount of smoke. However, it can be a source of model uncertainty, which is now also discussed in connection with the underestimation of modelled extinction profiles compared to lidar observations.*

*The study by Brown et al. (2020), mentioned by the reviewer, shows that aerosol-climate models (including ECHAM-HAM) may generally overestimate absorption by biomass burning aerosol due to an insufficient representation of the mixing state for fire aerosol. The revised version of the manuscript points out this uncertainty in more detail. For the extreme 2019/2020 wildfires, however, we can show in comparison with single scattering albedos (SSAs) derived from lidar measurements for this particular event that the assumptions about the optical properties of the smoke particles in the model are reasonable, with SSA values of 0.79–0.8 (550 nm) in the lidar inversions and slightly more reflecting values of 0.82–0.85 in the model.*

MC2) The Authors obtain a relatively large and positive RF at top of atmosphere (TOA), which is basically in contradiction with all estimations available at the moment (Khaykin et al., Yu et al., Hirsch and Koren, papers that are cited in the manuscript). The observational (Hirsch and Koren) and hybrid observational/modelling (Khaykin et al.) estimations agree on a relatively large negative TOA RF. The Authors interpret this disagreement with respect to these previous estimations as the result of all-sky calculations and the high surface reflectivity in the present manuscript, which, I agree, can partially explain that. By the way, the RF is very sensitive to optical properties of the aerosol layer (in particular, the absorption properties of the layer and its angular distribution of scattering, see discussion in SC56 and other SCs). In addition, also Yu et al. obtain a negative TOA RF but at all-sky conditions. This should be discussed more thoroughly in the text (see suggestions in several of the following specific comments) and the different statements supporting a positive RF must be smoothed a bit.

*The review of the above-mentioned papers shows that there is no reliable all-sky estimate of the aerosol radiative forcing for this event to be compared to our model results. They mostly refer to clear-sky conditions and predominantly to open ocean: (1) Khaykin et al. (2020) simply derive an all-sky forcing (RF) by assuming that "all-sky RF [are] reduced to about 50% of the clear-sky RF", which a priori excludes the possibility of a change in sign. (2) Hirsch and Koren present a clear-sky value from the CERES data that is at least significant at 20-60°S, but otherwise is not significant in the deviation from the CERES mean. However, they also find a*

*possibly significant positive forcing above clouds and Antarctica, but the low accuracy of the CERES data in these regions does not allow a more precise statement. (3) Yu et al. (2021) consider only clear-sky conditions but provide an effective forcing for shortwave and longwave radiation in contrast to the instantaneous forcing in this study. The atmospheric adjustments included in the effective forcing obviously reduce the instantaneous forcing considerably. Nevertheless, the instantaneous forcing we present is important as a measure of the energy added instantaneously to the atmosphere (here the stratosphere). Thermodynamical and dynamical adjusments are currently under research and will be treated more extensively in a future study.*

*Regarding the optical properties of the fire aerosol, the comparison with the lidar-based inversions of single scatter values was strengthened, and the asymmetry parameter was included in the discussion (see the reply to the specific comments). This analysis, however, further supports that the optical properties of the fire aerosol are reasonably realistic for this case, and thus the positive instantaneous solar radiative forcing at TOA.*

**Specific comments (#SC)**

SC1) P1 L24-25: "Global…wildfires": I would not call this "uncertainties" but rather "incomplete representation", like said in the following line, or something similar
*"show significant uncertainties" was replaced by "lack adequate descriptions of".*

SC2) P1 L27: Its more "observation-based input to the simulations" than "observation based approach"
*Agreed. Changed accordingly.*

SC3) P1 L28: please add "Based on our simulations,…" before "The 2019-2020 Australian fires caused…"
*Done.*

SC4) P1 L32: "While at surface,…" is an awkward way to start a sentence, please rephrase.
*Done.*

SC5) P1 L34: "deep wildfire plumes…", "deep" is a bit too generic here: do you mean "with high altitude injection" or just "extreme"?
*Exactly, high-altitude plumes were meant here. The previous wording was obviously inspired by deep pyro convection.*

SC6) P2 L5: "life": do you mean "wildlife"?
*No, we had all life in mind here, not just wildlife but also people. Therefore, we would prefer to keep just "life".*

SC7) P2 L6-7: "In addition…whether", this is an awkward sentence, please rephrase
*Agreed. The sentence was rephrased.*

SC8) P2 L14-17: please add SAGE-III and TROPOMI observations (as shown by Khaykin et al., 2020) into the discussion of evidences of the fire with satellites
*SAGE-III and TROPOMI were added to list of satellite detections.*

SC9) P2 L18-19: The first estimation of the radiative forcing of Australian fires was provided by Khaykin et al., 2020, with hybrid observations/modelling approaches. Please mention this manuscript and the method.
*The method and the estimates of the radiative forcing of the Australian fire aerosol by Khaykin et al. (2020) were added as requested.*

SC10) P3 L1-3: please break this very long sentence
*Done.*

SC11) P3 L5-6: Radiative-heating-induced self-rising can occur in fires but this is not the norm, so please smooth this sentence
*Agreed, the statement has been softened.*

SC12) P3 L9: "effects": Please specify which specific effects
*In order to be more specific, the sentence was adapted and now reads: "Such extreme wildfires and associated deep pyroconvection, for which injection of biomass burning smoke into the stratosphere has been observed, can have similar effects as volcanic eruptions in terms of stratospheric aerosol injection and radiative impact."*

SC13) P3 L10-11 "which is considered to be the strongest warming short-lived radiative forcing agent.": please add a reference for this statement
*In an earlier version of the manuscript, this referred to particulate climate forcers. As the current formulation is more general, we have corrected the statement and added references.*

SC14) P3 L11-12: "In addition…emitted": also precursors of secondary organic aerosols can be emitted, please mention
*Now also precursors of secondary organic aerosol are mentioned.*

SC15) P3 L12: "…radiative properties…": you might mean "optical properties"
*"radiative properties" replaced "optical properties".*

SC16) P3 L13-14: "…as well…altitude.": not clear, what do you mean?
*As explained by Ban-Weiss et al. (2012), absorbing aerosol like black carbon (BC) in lower atmospheric layers heats the surface due to diabatic heating. If located at higher altitudes, instead, it has a cooling effect, as the increased solar absorption is compensated by stronger outgoing long-wave radiation. Furthermore, Ban-Weiss et al. show that BC at high altitudes reduces high-altitude cloud cover, which also results in a surface cooling.*

SC17) P3 L16: "…the recent accumulation of extreme wildfires...": you mean "aggregated effect"?
*What was meant is "the recent series of extreme wildfires". The wording was changed accordingly.*

SC18) P3 L21: please remove "more"
*We believe "most" was meant, which we removed.*

SC19) P3 L27: not sure that "to capture" is the right verb here
*The verb "capture was replaced by "address".*

SC20) P4 L18: "height profiles": do you mean "vertical profiles"?
*Replaced.*

SC21) P5 L10-12: which altitudes for these vertical layers in the model?
*Note that these are not fixed injection heights. The PBL height is largely driven by the surface fluxes and associated vertical mixing. It therefore varies strongly in time and space. For this reason, no explicit altitude values can be given here.*

SC22) P5 L14: "47 levels": which approximate vertical resolution?
*The vertical resolution is variable across, but approximate values are now given for relevant altitudes: "In the vertical, the model is set up with 47 levels with increasing layer thickness from the ground to 0.01 hPa (~80 km). The vertical resolution ranges from approximately 70 m at surface to 500 m at 2.5 km and 1100 m at 15 km height and coarsens accordingly thereabove."*

SC23) P5 L18-19: "AOT and vertical profiles of extinction": This looks redundant as the AOT is just the vertical integration of the aerosol extinction.
*This may have looked redundant in the way it was written. However, the aerosol extinction profiles are calculated by an online lidar simulator that was implemented especially for such comparisons with CALIOP and ground-based lidar measurements. We have expanded the description.*

SC24) P5 L28: "Since no direct information was available on the actual pyroconvective injection heights...": This is not completely true as Khaykin et al. give an upper bound for the injection altitude at circa 17 km using CALIOP, and it is also shown at approximately this altitude for Hirsch and Koren, 2021 (as you mention later in the text). Please correct the sentence.
*This statement is meant literally. There was in fact no direct information at or above the wildfire sites, which can be typically used in models. There was no reliable radiative power information from satellites (otherwise it would have been included in the GFAS data) or other, possibly in-situ observations due to cloud cover (and heavy smoke). All approximate values for the injection heights in the studies mentioned are of course reasonable but based on satellite observations at some distance from the Australian continent or model assumptions from which the emission height was inferred. This is what we have also tested within the sensitivity study.*

SC25) P5 L37-38: The results of Hirsch and Koren (2021), seems more to show that smoke injection is injected at altitudes >16 km. Why do you say "14 km"?
*Hirsch and Koren (2021, suppl.) found "smoke fragments [...] located on the lower stratosphere below 17 km [...] during the fire emissions."*
*We find that the vertical spread in the CALIOP imagery justifies the "14 km" assumption, in particular, since due to the vertical resolution in the model in the tropopause region a larger altitude range is directly affected. We point this out more strongly in the text now.*

SC26) P5 L38-39: "In addition...the original biomass burning injection...": The original injection is what is described above (P5 L10-12)? Please clarify in the text.
*Yes. The details of the original fire injection are given here again to make this clear.*

SC27) Table 1: in the NoEmiss lines, when you say "1 April" you mean "4 January"?
*It is in fact 4 January. Many thanks for spotting this obvious mistake.*

SC28) More in general on the NoEmiss scenario: In the NoEmiss scenario, how are the previous emissions from Australian fires (i.e. the Australian fire season prior to 29/12/19) considered?
*All other days except the mentioned pyroCb days are treated as in the original configuration. This was added to the text for clarification.*

SC29) P6 L11: please suppress "significantly"
*Deleted.*

SC30) P6 L12: The long-term mean ("which implies…") is not shown in Fig. 1b, so please explain how it is calculated and rephrase the sentence?
*Please also note our response to reviewer #1. Now, Fig. 1b is already referred to in the introduction to show the hemispheric spread of Australian Smoke. Here, only the AVHRR AOT mean values for January are presented. The calculation is described in the Methods section.*

SC31) P6 L13: "AOT": please mention wavelength
*Done.*

SC32) P6 L14: "compared…half a year": this is not clear at all, please rephrase
*Deleted.*

SC33) P6 L15: "…relative to 2019": You mean wrt the monthly mean AOT for January 2019? Please clarify in the text
*Done.*

SC34) Figure 2: wrt MC1, the trend (maximum AOT in January and then decreasing) is not consistent with observations, e.g. the SAOD from SAGE-III in Khaykin et al., 2020 (Fig. 3). Please explain why
*Again, note that the peak AOT in Khaykin et al. (2020) is actually likely delayed. The authors point to saturation effects in the satellite retrievals in the caption of their Fig. 4 as an explanation.*

SC35) P7 L8: "The emissions…are reproduced…": The emissions are not "reproduced" by ECHAM but are "an input" to ECHAM: please rephrase
*We agree. The sentence was rephrased to read now: "The dispersal of this smoke plume is reproduced using the global aerosol-climate model ECHAM6.3-HAM2.3 with the pyroconvective injection heights prescribed."*

SC36) P7 L 10: "…provide an insight…due to wildfire smoke…": If NoEmis has only the smoke emissions of 29-31 December and 4 January switched off, then this comparison does not provide "the AOT distribution due to wildfire smoke" but rather "the AOT distribution due to pyro-convective events of 29-31/12 and 04/01". Please verify and possibly correct.
*Correct, this was well spotted and was corrected in the manuscript.*

SC37) P7 L16: "AOT differences": Differences with respect to what?
*This again refers to the difference between TP+1 and NoEmiss simulation results, which is now included in the text.*

SC38) P8 L15: "…is clearly better": Is it "clearly" better? Not to my eyes: the comparisons with different injection altitudes looks quite similar to me. This is not so surprising because the effect of the fires on the column AOT is not as strong as the one at selected UTLS altitudes (as visible in Fig. 4). Please, based on that, smooth these statements, and reconsider this discussion.
*The comparison of modeled and observed AOT in Fig. 3 may give this impression at first glance. However, considering the actual change in modeled AOT against the background of the in general very low levels of AOT at the Southern Hemisphere sites, the UTLS injection heights in fact lead to a substantial improvement. Now, we explicitly point out this fact.*

SC39) P8 L16-17: please suppress "using…above" (this is already clear from scenarios descriptions above, so is redundant)
*Deleted.*

SC40) P8 L18-19: "…indicating that the modeled effect…stratospheric smoke": This statement is not true because the solar absorption depends not only on the aerosol load but also on the optical properties of the aerosols - and then, for your estimations, on the assumptions made in the model on composition and atmospheric evolution of the smoke plume. Please smooth the statement.
*This statement was modified in the manuscript.*

SC41) P8 L19: "is larger" --> "is slightly larger"
*We disagree. The bias of the BASE case is on average about 30% larger than for the other cases using UTLS smoke injection. This quantitative information was added to the discussion.*

SC42) P8 L20: "…correlation is also lower": From Tab. 2 it looks like BASE R is quite comparable wrt TP(+-1) and the others setups.
*Replaced by "slightly lower".*

SC43) P8 L25-26: "which is also consistent with the CALIPSO satellite lidar observations": Which CALIPSO observations? (they're neither in Fig. 4 nor 5)
*The CALIOP comparison was deleted at this point.*

SC44) P8 L28: "reflect" is not a good choice here as a term, as it also has an optical meaning: please change term
*Replaced by "evident".*

SC45) P8 L28: "These remarkable values…": what do you mean?
*As now written, it is meant: "This remarkable wildfire smoke layering…shown in Fig. 4…"*

SC46) P9 L9-10: "again…apparent": You refer to Fig. 4 I guess: please mention this in the text.
*A reference to Fig. 4 was added.*

SC47) P9 L12-13: "The model results…smoke layer": This is not peculiar of the simulations but only empirically visible in Fig. 4 and 5 (for model as well as in the lidar observations): please rephrase.

*Agreed. We included a reference to the discussion of Figure 6 here.*

SC48) Figure 4: Please spell "coeff." and not "cf."; please use "km-1" as aerosol extinction units

*Corrected.*

SC49) Figure 5: here as well, please use km-1 as aerosol extinction units. Also, for the sake of visual clarity of the comparison, why not suppressing the pressure vertical axis and just put the height on the left, for your simulations results?

*The Figure was revised accordingly.*

SC50) P10 L10: "For other model scenarios": Please specify which scenarios.

*Thank you for pointing out this inaccuracy. Meant are of course only model scenarios with prescribed pyroCb smoke injection. This we added to the sentence.*

SC51): Your estimation of the heating rate: The heating rate is very sensitive to the aerosol optical properties, please mention this in the text as a reason for large uncertainties in your estimations. Also: are these shortwave-only or shortwave+longwave heating rates?

*Again, all estimates of smoke radiative effects in this study are for the solar wavelength band, which was made clear here also for the heating rates. The uncertainties due to the aerosol optical properties in the model are discussed in Sec. 3.2.*

SC52) Figure 6: are these "monthly averages" for January? Please mention this in the caption.

*Done.*

SC53) Figure 7: Would it be possible to have a altitude vertical axis as well?

*Figure 7 was completely revised. It now has a height axis. In addition, the geographical area was limited to the longitudes between Australia and Argentina (145°E - 70°W) and the tropics/subtropics (30°S - 60°S) were excluded as suggested by Reviewer #1. Furthermore, an averaging error in the previous version was corrected.*

SC54) P12 L6: "greenhouse forcing": You mean "greenhouse gases forcing"? Black carbon also can produce greenhouse effect (but it's particle, so better to be more specific)

*Correct, what is meant is the greenhouse effect caused by anthropogenic aerosols and gases, as it is now written.*

SC55) Your RF estimations: same question as for the heating rates: are these estimation for SW-only or LW+SW?

*All estimates of smoke aerosol forcing in this study are for the shortwave radiation. This was made clear in several places in the text.*

SC56) With reference to MC2: the RF of aerosols depends very strongly on the optical properties of the aerosol layer, which in turns, and this is very important for the complex

smoke emissions by fires, depend on the atmospheric evolution of the plumes. In particular, the aerosol RF depends quite strongly on both absorption properties (summarised by SSA) and the angular distribution of scattering (the phase function, summarised by the asymmetry parameter). Examples of such variability of RF on these two integral optical parameters (for volcanic aerosols, but it applies more in general) can be found here;- Sellitto et al., 2020 (https://www.nature.com/articles/s41598-020-71635-1), see their Fig. 5 - Kloss et al., 2021 (https://acp.copernicus.org/articles/21/535/2021/), see their Fig. 9. The situation can be even more complex for fire emissions, where the optical properties of the emitted and secondary formed aerosols, as well as their evolution in a complex environment of high humidity, many gaseous emission and locally high temperatures. Thus, your estimation depends strongly on the somewhat arbitrary assumptions of your simulations. This must be critically discussed in the text.

*The assumptions in this study are not arbitrary. The ECHAM-HAM model is a widely used community model that has been thoroughly evaluated for aerosol processes and aerosol-climate interactions. The only assumption that differs from the default configuration is the adjustment of the smoke injection heights for the 4 pyroCb days. This is also not arbitrary but based on ground-based and satellite observations and, moreover, is shown to be reasonably realistic by the evaluation performed.*

*In addition, the comparison of single scattering values in the model with the lidar-based inversions as well as the newly included mention of the asymmetry parameter shows that the particle optical properties are adequate for this fire event and thus the positive instantaneous solar radiative forcing at TOA.*

SC57) P12 L13-17: Yu et al. (2021) also obtain a slightly negative TOA RF using a model and theirs are all-sky estimations. Please notice that their estimations are SW+LW (personal communication). As a matter of fact, your RF being positive is quite in contradiction with all previous observations, simulations and hybrid estimations of the TOA RF for Australian fires 2019/20, clear- and all-sky, and this must be mentioned in the text.

*Yu et al. (2021) only look at clear-sky conditions. However, in contrast to the instantaneous forcing estimated presented in this study, they provide an effective forcing for shortwave and longwave radiation. The atmospheric adjustments included in the effective forcing reduces the instantaneous forcing, especially due to the high altitude of the smoke aerosol layer, resulting in an overall negative TOA forcing.*

*Note that our clear-sky instantaneous TOA forcing actually agrees well with the clear-sky effective TOA forcing from Yu et al. (2021) for solar radiation, for which the rapid adjustments are of minor importance.*

SC58) P12 L20-21: This is another reference that can be helpful in comparing your results with volcanic eruptions: https://www.nature.com/articles/ncomms8692?proof=t
*Many thanks for this additional reference.*

SC59) P12 L22-24: No mention to the stratospheric vortex driven by rapid vertical transport + plume heating seen for this fire event? (Khaykin et al., 2020; Kablick et al., 2020). *This was implicitly meant by the "responses in atmospheric dynamics". However, this is now explicitly mentioned: "Khaykin et al. (2020) actually showed that a self-sustained 1000-km anticyclonic vortex formed as a result, which traveled through the stratosphere for weeks, accompanied by a local ozone reduction".*

SC60) P13 L7-8: "uncertainties in AOT; in particular…single scattering albedo…": This should be rephrased: the AOT representation and single scattering albedo are only in part inter-dependent. In addition, the angular scattering properties of the aerosol layer is also (or, at some conditions, even more) important for RF estimations (see Sellitto et al., 2020; Kloss et al. 2021, mentioned at SC56), and this should be mentioned.

*This sentence now includes the asymmetry parameter as a source of uncertainty, which is additionally addressed later in this section. See also the response below.*

SC61) P13 L9: "SSA lies between 0.82-0.85…". Single scattering albedo (and asymmetry parameter of soot aerosols, see SC60) may be significantly affected by their mixing states, and coating of BC. SSA can be significantly larger (up to ~0.95 at 550 nm) if BC is coated by aqueous secondary aerosols (organic or sulphate) - e.g. https://www.nature.com/articles/s41467-020-20482-9 . Also, and importantly, it looks like smoke aerosols are too absorbing in models due to a generally incomplete representation of the aerosol mixing state for biomass burning aerosols: https://www.nature.com/articles/s41467-020-20482-9 . This must be mentioned and discussed in the text, as this might be a large source of uncertainties for your RF estimations (as well as heating rates estimations and, even more at the basis, AOT fields).

*We are aware of this study on the apparently widespread overestimation of absorption of biomass burning aerosol in aerosol-climate models.*
*The revised version of the manuscript now discusses this point in more detail, including a reference to the study of Brown et al. (2020). In addition, the comparison with current lidar-based derivations of SSA values for other intense fire events and for this one in particular was made more concrete. These values, however, support our previous conclusion that smoke absorption is even slightly underestimated in the model for this extreme wildfire case.*
*Regarding the asymmetry parameter, it is difficult to make an evaluation because the exact morphology of the smoke particles is not known. For the smoke particles, only the asymmetry parameter of the fine mode fraction is relevant here. In our model, the asymmetry parameter for the Australian smoke is about 0.6 at 550 nm, which however is in good agreement with typical values for wildfire aerosol in the literature (see e.g., Reid et al., ACP, 2005).*
*In the case of the volcanic ash, mentioned by the Reviewer, the asymmetry factor is even more uncertain due to the more irregular particle shapes (depending on the eruption type) and the uncertainties in radiative forcing may not be directly comparable.*

SC62) P13 L11-12: "For the 2019-2020 Australian fires, first inversion results even point towards SSA values just below 0.8, as calculated using the method of Veselovskii et al. (2002).": more details are called here.
*The sentence was rephrased to include more details as well as a reference for the lidar-derived SSA value.*

SC63) P13 L14-15: "Thus, together with the low bias of the modeled smoke AOT, we argue that our results illustrate a conservative estimate for the positive TOA forcing of this event.": This statement is definitely too strong, due to the large uncertainties discussed in my previous comments and to the fact that all previous RF estimations (Khaykin et al., Hirsch and Koren, Yu et al.) indicate rather a negative RF at TOA.
*Based on the revised discussion on the uncertainty factors of the model estimates of the radiative forcing for the Australian smoke, we see us actually more confirmed in our statement (see the response to SC61). It should also be emphasized again that this study, unlike those*

*mentioned, provides an estimate of the radiative forcing of smoke for all-sky conditions as well as for snow and ice-covered areas over Antarctica, not only predominantly over open ocean. The assumption for all-sky conditions in Khaykin et al. is an approximation rather than a modeled estimate ("all-sky RF reduced to about 50% of the clear-sky RF"). Yu et al. consider only clear-sky conditions. And Hirsch and Koren find that "the SW all-sky [...] deviations from the [CERES] mean are not consistent throughout the whole period".*

SC64) Table 3. As already said for the AOT trend, in Khaykin et al. the strongest RF is in February and here is in January. As already mentioned in previous comments, this might be linked to an insufficient representation of secondary aerosols formation in your model and mixing state. Please mention and comment in the text.
*As already stated in reply to comments MC2 and SC56, this comment originates from overlooking the saturation effect in the satellite measurements mentioned by Khaykin in the figure caption of Fig. 4, which explains the delayed aerosol maximum in the observations. Nevertheless, it is possible that in this study the model underestimated the secondary aerosol formation, however, this would not have such large extent. We mention this as a further potential source of underestimated extinction in the revised manuscript.*

SC65) P14 L15: "While this is appropriate…pyroCb clouds": Thus, why not using the satellite observations of the plumes themselves as proposed by Kloss et al. (2021, Fig. A1-2), see also SC56?
*It is certainly a good idea to further explore the potential of satellite observation for initializing the injection heights of vegetation fires. However, bush or forest fires are more complex than volcanic eruptions as described in the mentioned paper. They are spatially variable and can extend over larger areas, which is also difficult to predict precisely. For modelling applications, satellite-based radiative power has proven to be a good measure of the height of smoke injection into the atmosphere, which can be well attributed to individual fire sites. However, it has been shown that, just as in this case, heavy smoke and cloud development strongly hamper detection.*

SC66) P14 L20-21: "Consequently, aerosol-climate models underestimate the wildfire aerosol impacts on the energy balance, as the vertical location of the smoke relative to clouds is fundamental to its radiative impact.": This might be true for pyro-convective fires while it is demonstrated that the biomass burning RF is overestimated in general, in models, again available at the following link:
https://www.nature.com/articles/s41467-020-20482-9
*Agreed. The statement was made more precise.*

SC67) P14 L29: please put references in chronological order.
*Done.*

---

## Author Response (AR2)

**Authors' Response to Reviewers' Comments**

Manuscript No.:   acp-2021-862, submitted to GMD
Title:            Important role of stratospheric injection height for the distribution and radiative forcing of smoke aerosol from the 2019/2020 Australian wildfires

Authors:          Bernd Heinold, Holger Baars, Boris Barja, Matthew Christensen, Anne Kubin, Kevin Ohneiser, Kerstin Schepanski, Nick Schutgens, Fabian Senf, Roland Schrödner, Diego Villanueva, and Ina Tegen

**Anonymous Referee #1 (RC1), 2nd round**

Compared to the first version the paper improved a lot. But there are still some minor issues to be corrected.

*We sincerely thank the anonymous reviewer for his time and constructive comments also in the second round. We hope that we have responded satisfactorily to all the questions and that any remaining unclear points could be clarified.*

**Specific comments**
(The line numbers refer to the version with change tracking.)

Page 4, line 18: I wonder why there is no routine similar to the one for the lidars (page 6) for output sampling at the AERONET stations. It is not necessary to repeat calculations.
*The standard output of ECHAM-HAM are AOD values at 550 nm, 865 nm and 55,555 nm. 550 nm is a common wavelength for satellite products, but also some AERONET stations provide values for this wavelength. The lidar simulator, on the other hand, is a relatively new implementation, specifically for CALIOP comparisons, and therefore directly provides the optical parameters at 532 nm. Since AERONET observations are often used for evaluation, it is certainly useful to extend the default output in a future model version. Nevertheless, the conversion using an Angstrom parameter is a suitable as well as common method.*

Page 5, line 22ff: Is this also the case for the lower stratosphere? The statement appears to be not consistent to some results shown in Figure 7. Do the 44% refer to the total column or some layers?
*Yes, indeed, this statement also applies to the lower stratosphere. Thank you for pointing this out, we have added it accordingly and now also include a reference to Liu et al (2019) who evaluated the performance of the CALIOP V4 CAD algorithm in the troposphere and stratosphere: "While the V4 CAD can distinguish aerosols and clouds for stratospheric layers, uncertainties tend to increase as the altitude increases. This increasing uncertainty derives from the fact that the very low aerosol occurrence frequency at high altitudes does not provide a statistically significant sample size to constrain the [probability density functions] PDFs [...]".*
*However, we agree that in this particular case the discrepancy between modeled extinction coefficients and those observed by CALIOP decreases again in the uppermost part of the*

*profile (>23 km). Although it seems that there the CALIOP profiles for all years converge to some background value.*
*Here, "globally" is meant literally. The 44% value is a vertical mean across the entire globe (Watson-Parris et al., 2018).*

*Additional reference:*
*Liu, Z., Kar, J., Zeng, S., Tackett, J., Vaughan, M., Avery, M., Pelon, J., Getzewich, B., Lee, K.-P., Magill, B., Omar, A., Lucker, P., Trepte, C., and Winker, D.: Discriminating between clouds and aerosols in the CALIOP version 4.1 data products, Atmos. Meas. Tech., 12, 703–734, https://doi.org/10.5194/amt-12-703-2019, 2019.*

Section 2.4: The dual call of the radiation code for calculation of forcing and heating is not mentioned here in contrast to the reply to the referees. At least it is mentioned on page 13.
*That's right. Eventually, we decided that the description is best placed where the instantaneous forcing is actually discussed. However, additional details on the radiation scheme in ECHAM-HAM, including the dual call for diagnosing the instantaneous aerosol radiative forcing, are now given in Section 2.4.*

Page 6, line 30, abstract and introduction: Is the agreement of mass by chance or are the data taken from the references?
*No. As we describe in this line and several times in the text, the strength of the biomass burning emissions in the model is based exclusively on the GFAS product and were not further adjusted for the 2019/2020 Australian fires. An adjustment was only applied with respect to the injection height as described in Section 2.4.2. This shows the quality of the GFAS emission inventory. For clarification we add: "These values agree well with the previously mentioned estimates by Peterson et al. (2021) and are kept unchanged throughout the sensitivity experiments."*

Page 6, line 36f and Table 1: Is there a gap between the layers with smoke in the scenarios TP1_8020 and TP1_5050 or do you mean the layer containing the tropopause instead the one below the tropopause? More precise please. I suppose you mean "all wildfire smoke" in TP+1, if yes please include in Table 1.
*There is of course no gap between layers with smoke in the scenarios TP1_8020 and TP1_5050. To be more precise the sentence was modified as follows "TP1_8020: as TP+1 but only 80% of the emitted smoke injected above the tropopause and 20% distributed between tropopause level and surface."*
*In the sentence above we clearly say that only the emission heights over Southeastern Australia were modified.*

Page 10, line 23: Also scenario NoEmiss, right?
*That's true. Here, the misrepresentation of the stratospheric injection height has the same effect as if the fire emissions of the pyroCb days had not been considered at all. We add: "…just like in the NoEmiss case."*

Page 11, line 2: Mention Fig. 6.
*Done.*

Figure 6: "DRF" in the caption is not defined, please spell out somewhere.

*Done.*

Page 12, line 15ff: It might be good to mention a typical range of the lidar ratio used by the simulator to create Fig.7.
*In order to produce Fig. 7, which shows extinction profiles, no assumptions about the lidar ratio are necessary on the model side (lidar simulator). As a matter of interest and for evaluation of the smoke optical properties in the model, Section 3.2 now includes a statement as follows: "Accordingly, Ohneiser et al. (2020, 2022) report lidar ratios at 532 nm in the range of 75 – 112 sr (average 97 sr) for their Punta Arenas observations, also indicating strongly absorbing aerosol. The lidar simulator of the model, in comparison, provides slightly lower lidar ratios at 532 nm between 70 and 100 sr for the stratospheric smoke layer."*

Page 14, line 14: Do you see similar features in your simulations? You should include a sentence on that even if it is modified by nudging.
*Yes, such a vortex is also seen in the 50-hPa wind fields of the ECHAM6.3-HAM2.3 simulations. However, the analysis of the atmospheric dynamic effects is the subject of a follow-up study. A sentence to this effect was added.*

**Technical corrections**

Page 3, line 5: Correct grammar.
*Done.*

Page 4, line 15, line 30 and Page 6, line 16f: Are the calculations of optical properties really done for the three wavelengths 532, 550 and 553nm or extrapolated like stated on page 4?
*The wavelength of 553 nm for the lidar simulator is a typo. The calculation of the 550-nm optical thickness using the Angstrom parameter is only done for the AERONET observations. The lidar simulator provides aerosol extinction profiles at 532 nm that can be used directly for comparisons with CALIOP and the ground-based lidar at Punta Arenas.*
*Thank you for pointing this out.*

Page 6, line 41: "by" instead of "due to"?
*"By" does not express the explanation that is meant here. Still replaced by "because of".*

Table 1: I suppose you mean "the layer around 14km".
*Corrected.*

Figure 2: Please check the color code. I don't see the high values stated in the text in the figure.
*Thank you for bringing this inconsistency to our attention again. There is a misunderstanding here that we overlooked in the last round of reviews. In the text, monthly means of the AOT of the respective scenarios (BASE, TP+1) are given in absolute values. In contrast, Figure 2 shows the differences in mean AOT for January to March 2020 between the model scenarios BASE (2a-c) and TP+1 (2d-f) against the NoEmiss results, respectively. Therefore, they exclusively show the contribution of the smoke-only AOT for the case where no smoke injection by pyroconvection is prescribed in the model (Fig. 2a-c) or for smoke injection into the model layer above the tropopause on the AOT for pyroCb days 29-31 December 2019 and 4 January 2020 in southeastern Australia (Fig. 2d-f).*

*In order to clarify, the sentence was modified as follows: "While the monthly mean smoke AOT is simulated in absolute values as high as 0.26 and 0.22 for January just downwind of the fire region in Southeast Australia for the BASE and TP+1 experiments, respectively,…"*

Page 20, line 9: You should include a doi for that online reference.
*We are not sure which of two references is meant, but both Heinold et al. and Hersbach et al. include the doi in a format suggested by Zenodo and RMetSoc, respectively.*

**Anonymous Referee #2 (RC2), 2nd round**

I would like to sincerely thank the Authors for their work and revisions of their very interesting manuscript. I can see that most of my Specific Comments have been tackled and I'm very satisfied of the way this was addressed by the Authors. By the way, my main concerns have been mostly skipped, namely the model's representation of secondary aerosols (Major Comment 1 and a few Specific Comments) and the role of evolving optical properties on the radiative impacts (Major Comment 2 and a few Specific Comments). To be clear, I honestly think that the paper should be published soon, as it deals with an important topic, but I would in any case require that these two points are better addressed before publication. This basically means: 1) smoothing many very strong statements (e.g. about the "certainly positive radiative forcing of the plume" or the "perfect optical properties simulated by the model" or the "secondary aerosols which are surely not formed") and 2) adding a deeper and comprehensive discussion on the two issues. I strongly suggest the Authors to make this effort. A few more details are in the following.

Thank you for the interesting work and discussion,
Pasquale Sellitto

*We also thank you for your availability for the second round of review and the critical discussion and suggestions that helped to further improve this work.*

*As outlined in the response to the comments below, we have tried to address your comments as best as we could. Specifically, we have included secondarily formed particles as a possible reason for the underestimation of the modeled smoke aerosol. In general, we point more strongly to the uncertainties in the optical properties in our model results. And we discuss the possibility that dilution impacts on smoke aging could have led to temporal and spatial variability in smoke optical properties, which cause further uncertainties in the model estimate of the direct radiative forcing of the 2019-2020 Australian wildfire smoke.*

**Major comments (#MC)**
**MC1)** In Khaykin et al., 2020, it was supposed that one reason for the increasing trend of SAOD could be saturation of the OMPS-LP detector. This was the proposed reason at the time of publication of that paper (which I personally co-authored). By the way, following reflections and analyses since that publication, while keeping this as a possible explanation of this time evolution of the SAOD, led to other possible explanations:

1) the formation of secondary aerosol and aerosol mixing, (which, on the other hand is a known issue in terms of representation of biomass burning plumes in models like the one used in this work (Brown et al., 2020)) and 2) plume's progressive lofting due to in-plume radiative heating. These aspects are further addressed and discussed in a recent preprint publication (https://egusphere.copernicus.org/preprints/egusphere-2022-42/) that I suggest checking. As a matter of fact, these two aspects cannot be excluded, and this certainly needs further discussion in the paper. Why are you so deterministic in this statement "Secondary aerosol formation appears unlikely to be the explanation considering the required amount of smoke."? The progressively less absorbing aerosol properties seem to actually point at a progressive secondary aerosol formation and brown-carbonification of black carbon emissions.

*Thank you for bringing this follow-up paper to our attention. In particular, the sensitivity study is very interesting with respect to the effects of plume aging and mixing on the evolution of smoke optical properties and ultimately the radiative forcing. We picked this up now, where it fit.*
*In reply to your comment, we would first like to emphasize that we already pointed out several times in the last paper version the overestimation of smoke aerosol absorption in this model and climate models in general, as noted by Brown et al. (2020), resulting from an inadequate representation of the plume aging and particle mixing state. And we already discussed (page 10, line 17/18 of the new change-tracked version) that "The underestimation of the fire aerosol loading in all configurations […] is partly due to missing secondary aerosol formed in the plume, which is not considered by the model." Just as you say, the heating-induced lofting affects the radiative influence of the smoke plume. This effect is also reproduced by the model as we show in Section 3.1.*
*So, we are in fact aware of the uncertainties due to secondary aerosol formation and the evolution of fire aerosol properties. On the other hand, we cannot ignore the good agreement of the model results with the lidar measurements and retrievals of optical particle properties from southern Chile, which apply very well at least for this particular pyroconvective case. These lidar data are the best constraints of aerosol optical properties available for this Southern Hemisphere wildfire event.*
*In contrast, the problem of the delayed SAOD peak in February, which could be due to saturation of the OMPS LP detector, does not seem to be resolved in your study* (*https://egusphere.copernicus.org/preprints/egusphere-2022-42/*): „… we cannot exclude any of the above hypotheses, we are inclined to consider the aging of the plume as an important factor at play...". One of the EGUsphere reviewers also points out this possible weakness of the instrument. We cannot judge this, but tend to believe the measurements from AERONET and the lidar observations at Punta Arenas, which point to a peak already in January 2020.*
*Nevertheless, we think you raise an important point here regarding the temporal and spatial evolution of the plume optical properties, especially with respect to secondary aerosol formation and mixing. So, we mention secondarily formed particles as a possible reason for the model underestimation of fire smoke more often in the text, and now also refer more strongly to the uncertainties in the optical properties due to secondary fire aerosol and aging that might have played a more important role on the larger scale in Section 3.2 (further details below).*

The LiDAR SSA inversions (by the way, please discuss briefly the inversion methodology in the Data and Methods section) that are now presented in the manuscript cannot actually demonstrate the fact that there is no secondary aerosol formation, and then progressive larger SSA and lesser absorption from plume's aerosol, because: 1) if I got it right, these are measurements for January 2020 only, too early to have a marked secondary aerosol formed and a clear signature in the plume's aerosols optical properties; 2) LiDAR inversions of optical properties have usually significant uncertainties and SSA variability is small (from 0.75-0.80, for black carbon; 0.85-0.90, for brown carbon, only 10-15% increase on SSA but sufficient to switch the RF sign from positive to negative). Also, the statement, P15 L3-4: "Ohneiser et al. (2022) show an SSA of 0.79 for the rotating smoke disk on 26 January above Punta Arenas in Chile, which is also representative for other smoke measurements" is not true: the vortex plume is an isolated patch of fresh smoke aerosols, isolated from the environment and absolutely not representative, in terms of optical properties, of the overall large-scale plume: please correct. It is necessary that you add a substantial discussion on these aspects in your manuscript and be more cautious in this respect in the Abstract and Conclusions.

*It was only in January that the smoke AOT was high enough for Ohneiser et al. (2022) to perform the multi-wavelength inversion to derive the single scattering albedo from the Polly lidar measurements in Punta Arenas/Chile. At this time, the SSA was about 0.80 with an error of about 0.05. As Ohneiser et al. (2022) show (see their Figure 8b), the 532 nm lidar ratio remained high well beyond the end of January, indicating strongly absorbing aerosol, which is why low SSA values can be assumed to continue occurring in February and March. Regarding the formation of secondary aerosol and aerosol mixing, we think that the condensation of gases onto smoke particles usually lasts on the order of 2 days in the troposphere, while it will certainly continue for longer durations in the stratosphere. The observed decline in the lidar depolarization ratio at Punta Arenas marks the completion of the aging of the smoke aerosol by mid-February at the latest (see Fig. 8c in Ohneiser et al., 2022), with low depolarization values indicating aged round particles after this time. After the condensation phase, only coagulation is considered to change the smoke size distribution, and coagulation is not as effective in the stratosphere as in tropospheric layers.*

*The statement that the smoke optical properties obtained for January 2020 are also representative for other mid and high latitudes in the Southern Hemisphere is actually a conclusion by Ohneiser et al. (2022) that we refer to. We agree that the vortex structure most likely preserved the enclosed smoke plume from being rapidly diluted within the environment. However, there is probably no reason why the smoke trapped in the vortex should not be subject to aging or SOA formation, with volatile precursors being co-emitted with BC and coagulation also taking place. On the other hand, it cannot be excluded that dilution impacts on the aging of the plume may have led to a different evolution of the smoke optical properties between the vortex and larger scale environment, which were not shown by the lidar measurements.*

*Accordingly, the discussion in Section 3.2 was revised and extended, including more references to the findings by Ohneiser et al. (2022). Furthermore, we included secondarily formed particles as a possible reason for the underestimation of the modeled smoke aerosol, for example in Section 3.1. We also point more strongly to potential uncertainties in the optical properties in our model results. In addition, the possibility is discussed, that dilution*

*effects at the plume edges in contrast to the core could have influenced the spatio-temporal evolution and variability of the smoke optical properties, which may not have been captured by the local lidar observations but may still be a source of uncertainty in model estimates of radiative forcing.*

*The lidar inversions of single scattering albedo (SSA) were not performed in this study but are actually part of the work by Ohneiser et al. (2022), which we refer to in the discussion in Section 3.2. Going into details would be beyond the scope of this paper. Nevertheless, it is certainly a good idea to refer to the inversion method by Veselovskii et al. (2002) and the study by Ohneiser et al. (2022) already in the Data and Methods section.*

**MC2)** First, please accept my apologies for my mistake: Yu et al. (2021) is also clear-sky RF estimations and not full-sky as I stated in the previous review round. By the way, it is undoubtedly true that optical properties of the aerosol layer have dramatic impacts on the radiative forcing of a given aerosol layer, which is even more important for biomass burning highly evolving plumes. The LiDAR observations and all discussion in the revised manuscript only deal with the young plume (in January), while the optical properties of biomass burning aerosols should evolve (e.g. SSA and g) at longer timescales and mostly visible, in case, starting from February-March. Thus, it cannot be accepted what you state: "This analysis, however, further supports that the optical properties of the fire aerosol are reasonably realistic for this case, and thus the positive instantaneous solar radiative forcing at TOA". Again, yours is a valuable work and should be published but the limits of the model assumptions must be discussed, and the fact that the magnitude and sign of the radiative forcing depend on the modelled aerosol optical properties must be clearly stated. The strict certainty of a positive radiative forcing, that you suggest, should be avoided throughout the whole text. In the meanwhile, a preprint with sensitivity analyses of radiative forcing for this event to optical properties has been published (see MC1); please exploit, in your paper, these sensitivity analyses in the discussion of this aspect.

*As described above, the Ohneiser et al. (2022) observations support the assumption that the aging process of the Australian fire plume was completed by the end of January, but that the smoke particles continued to be absorbing as indicated by persistently high lidar ratios. Since the previous paper version apparently gave the wrong impression that we were referring only to the lidar observations in January, we expanded Section 3.2 for clarification, as already replied to MC#1.*

*We were happy to include your study in the introduction and discussion. And we point more strongly to the uncertainties in the optical properties in our model results, as well as the possibility that this secondary fire aerosol and aging have had a larger effect on the larger scale. Please note in this regard our response to MC# 1 and, in particular, the modifications to the text in Section 3.2 and Section 4 'Implications and perspectives'.*

---

## Author Response (AR3)

**Authors' Response to Reviewers' Comments**

Manuscript No.: acp-2021-862, submitted to GMD
Title:            Important role of stratospheric injection height for the distribution and
                 radiative forcing of smoke aerosol from the 2019/2020 Australian wildfires

Authors:         Bernd Heinold, Holger Baars, Boris Barja, Matthew Christensen, Anne
                 Kubin, Kevin Ohneiser, Kerstin Schepanski, Nick Schutgens, Fabian Senf,
                 Roland Schrödner, Diego Villanueva, and Ina Tegen

*We sincerely thank the anonymous reviewer for his time and constructive comments also in the third round. With this revision, we think to have addressed all remaining comments.*

**Anonymous Referee #1 (RC1), 3nd round**

I would recommend publication after minor corrections:

**Minor comments**
(line numbers refer to the revised manuscript with tracked changes ATC2)

Page 1, line 37: You may insert "which is consistent with our all-sky values" or something similar since the assumption for global average cloudy sky is rather artificial.
*Our estimates of all-sky forcing are based on modelled but reasonably realistic land type and cloud distributions in the aerosol-climate model ECHAM-HAM. In contrast, Sellitto et al. (2022) assume a surface reflectivity of 0.5 globally to represent the cloud effect. It is therefore not fair to write that the values are consistent. In fact, their assumptions are rather artificial, which is why we use again the term equivalent in the sentence: "[…] calculated a global equivalent of TOA forcing as high as […]".*

Page 1, line 38: You may insert after "1.1W/m2" for clarity "(i.e. negative forcing)"
*Good point, done.*

Page 6, line 8: The references Iacono et al., 2008 and Pincus and Stevens, 2013 are missing in the reference list (Page 18ff).
*The missing publications were added to the reference list.*

Page 21, line 25: Please include coauthors and doi in the Jolly-reference.
*Done.*

**Referee #2 (RC2), 3nd round**

I sincerely thank again the Authors for their revisions and the very interesting discussion that originates from it which, it is important to notice it here, is an integral part of the ACPD review process and of the general journal philosophy. I think that such discussions between modelers and experimentalists is quite necessary in our scientific domain.

In general, I'm very satisfied of how most of the strong statements have been smoothed in this version. I see that a few divergences still are there, between the Author's view and mine on the subjects of: 1) the aerosol extinction and absorptivity evolution during plume's dispersion, and 2) the representativity of the point LiDAR observations at Punta Arenas. For the sake of the discussion, I will list my further concerns in the following but leave it up to the Editor to see if more modification of the text is needed. I also add a list of specific corrections, which are more recommended.

Apart from that, I recommend the manuscript for publication on ACP and I warmly thank the Authors for the great paper.
Pasquale Sellitto

*Again, we thank you for your availability also for the third round of review. We acknowledge the critical and fruitful discussion within the review process of this study, where we have definitely learned a couple of things from the perspective of the volcanic research.*
*We have tried to make further improvements and caveats as we could follow the arguments and hope to have found good counter-arguments for the others.*

**Outstanding divergences**
1) The Authors insist on the fact that the ageing processes of the plume are over by end of January, also based on what discussed in Ohneiser et al., 2022. The buildup of secondary aerosols in the stratosphere can take several weeks (see also volcanic plumes in the stratosphere) and the ageing processes of biomass burning aerosols can be even more complicated and take longer. From a theoretical point of view, I don't see any reason to affirm that this is over by January. A marked increase in the AOD is observed and available in the literature for different biomass burning plumes at the temporal scale of up to some weeks. In Ohneiser et al., it can be seen that depolarisation can be significantly larger than a few percent (up to 10%) until end February/beginning March, which does not support the idea of a plume ageing already finished by January.

*We strongly disagree. This study does not say that plume ageing will be completed by the end of January. Based on Ohneiser et al. (2022), we refer to the temporal evolution of the lidar depolarization ratio at Punta Arenas, which clearly indicates the ageing of the plume by mid-February already, with a decrease from 20% to 10%. We have further clarified this to mention also the further reduction of the depolarization ratio to 5% (and ongoing ageing) after the end of February. This, however, does not affect our reasoning.*

There is a decline in AOD at Punta Arenas, this is true, but how sure are the Authors that this station is truly representative of the whole SH mid and high latitudes, as stated in the text? Thus, I would recommend (up to the Editor to see if this is truly necessary) to add a sentence at P8, after L10-11, to mention that in any case the present modelling results in terms of the

AOD trend, even if in agreement with a single LiDAR station, are in disagreement with limb satellite observations, like those shown in Khaykin et al. 2020 and Sellitto et al., 2022 (which show an AOD increase from January to February). Possibly, also the statements about the representativity of the single point LiDAR observations at Punta Arenas should be smoothed.

*Unfortunately, there was a misunderstanding. We wrote in plural that the ground observations peaked around or after mid-January. This was not limited to Punta Arenas but included the other southern hemisphere AERONET stations. We have therefore added 'AERONET' here. By the way, this does not refer to the Polly lidar, but to the AERONET sun photometer measurements in Punta Arenas.*
*Because of the possibly unclear saturation issue of the mentioned OMPS LP satellite instrument and the related potential delay in smoke detection (see Khaykin et al., 2020) we would prefer not to include the sentence as suggested.*

2) The Authors bring the consistently large values of LiDAR ratio (>70 sr) as an indication of the fact that aerosols in the plume stay absorbing until March. The LiDAR ratio depends on the aerosol composition but also on the size of the particles. While BC aerosols have usually a larger LiDAR ratio value with respect to other aerosol types, the LiDAR ratio has also tendency to increase with increasing aerosols mean size. **As aerosols are probably increasing in size due to ageing, the fact that the LiDAR ratio stays large until March simply cannot be used as an argument to infer an aerosol large absorptivity. As a matter of fact, LiDAR ratios as large as >80 sr have been found even for the purely scattering sulphate aerosols of some extreme volcanic eruptions** (see e.g. Prata, A. T., Young, S. A., Siems, S. T., and Manton, M. J.: Lidar ratios of stratospheric volcanic ash and sulfate aerosols retrieved from CALIOP measurements, Atmos. Chem. Phys., 17, 8599–8618, https://doi.org/10.5194/acp-17-8599-2017, 2017). Thus, I would recommend (up to the Editor to see if this is truly necessary) to **modify or even suppress the sentence at P15 L1-3: "while high lidar ratios…Ohneiser et al., 2022)**".

*The lidar ratio is greatly affected by the particle properties. However, the analysis by Cao et al. (2019) shows that for the particle size range here, there is only an unnoticeable increase in the lidar ratio with particle radius.*
*On the other hand, the mentioned study by Prata et al. (2017), argues that the partially quite high lidar ratios for some volcanic ash plumes can rather be explained by different particle populations or types (sulphate-rich versus ash-rich aerosol layers) that they have been able to discriminate.*
*Therefore, we keep considering our argument that more or less constant high lidar ratios of approximately 91 sr on average (!) point to particles that continue to be absorbing.*

*References:*
*Cao, N., Yang, S., Cao, S. et al. Accuracy calculation for lidar ratio and aerosol size distribution by dual-wavelength lidar. Appl. Phys. A 125, 590 (2019). https://doi.org/10.1007/s00339-019-2819-y.*

*Prata, A. T., Young, S. A., Siems, S. T., and Manton, M. J.: Lidar ratios of stratospheric volcanic ash and sulfate aerosols retrieved from CALIOP measurements, Atmos. Chem. Phys., 17, 8599–8618, https://doi.org/10.5194/acp-17-8599-2017, 2017*

**Specific corrections**

1) L33-34: "...and subject to the model uncertainties in the smoke optical properties": this is said to apply to the surface RF but in fact this also applies to top-of-atmosphere RF (discussed at L30-32). Please correct the phrasing to account for that.
*Done.*

2) P5, L5-6: "In addition, values...data inversion (Ohneiser et al., 2022)": please specify that the single scattering albedo is specifically retrieved in that paper only for January
*Done.*

3) P15, L10-11: "In terms of single scattering albedo...": you cannot use SSA observations with this LiDAR to judge your modelling estimations quantitatively because you strictly have SSA retrievals only for January. Please correct.
*The statement was limited to January 2020, for which the retrieval is actually available. Below in the text, we had already pointed out during the last revision that a larger contribution to stratospheric fire aerosol cannot be ruled out and that dilution effects on the spatio-temporal evolution of the smoke plume may not or only insufficiently be represented.*